# Coral-associated bacteria demonstrate phylosymbiosis and cophylogeny

F. Joseph Pollock[1], Ryan McMinds [2], Styles Smith[1], David G. Bourne[3,4], Bette L. Willis[3,5], Mónica Medina [1,6], Rebecca Vega Thurber[2] & Jesse R. Zaneveld[7]

Scleractinian corals' microbial symbionts influence host health, yet how coral microbiomes assembled over evolution is not well understood. We survey bacterial and archaeal communities in phylogenetically diverse Australian corals representing more than 425 million years of diversification. We show that coral microbiomes are anatomically compartmentalized in both modern microbial ecology and evolutionary assembly. Coral mucus, tissue, and skeleton microbiomes differ in microbial community composition, richness, and response to host vs. environmental drivers. We also find evidence of coral-microbe phylosymbiosis, in which coral microbiome composition and richness reflect coral phylogeny. Surprisingly, the coral skeleton represents the most biodiverse coral microbiome, and also shows the strongest evidence of phylosymbiosis. Interactions between bacterial and coral phylogeny significantly influence the abundance of four groups of bacteria–including *Endozoicomonas*-like bacteria, which divide into host-generalist and host-specific subclades. Together these results trace microbial symbiosis across anatomy during the evolution of a basal animal lineage.

[1] Department of Biology, Pennsylvania State University, 208 Mueller Lab, University Park, State College, PA 16802, USA. [2] Department of Microbiology, Oregon State University, 226 Nash Hall, Corvallis, OR 97331, USA. [3] College of Science and Engineering, James Cook University, Townsville, QLD 4811, Australia. [4] Australian Institute of Marine Science, Townsville, QLD 4810, Australia. [5] ARC Centre of Excellence for Coral Reef Studies, James Cook University, Townsville, QLD 4811, Australia. [6] Smithsonian Tropical Research Institute, Smithsonian Institution, 9100 Panama City PL, Washington, DC 20521, USA. [7] Division of Biological Sciences, Bothell, School of Science, Technology, Engineering, and Mathematics, University of Washington, UWBB-277, Bothell, WA 98011, USA. These authors contributed equally: F. Joseph Pollock, Ryan McMinds. Correspondence and requests for materials should be addressed to J.R.Z. (email: zaneveld@gmail.com)

Since their first appearance around 425 million years ago, scleractinian ('stony') corals (Cnidaria: Hexacorallia: Scleractinia) have radiated into more than 1500 species, many of which serve as the major architects of coral reef ecosystems worldwide[1]. Modern corals harbor complex communities of microorganisms, including dinoflagellates, fungi, bacteria, and archaea which are collectively termed the coral microbiome[2]. Shifts in the composition of the coral microbiome and virome are linked to changes in coral health, disease, and resistance to stressors[3–6]. It is likely that ancestral corals also harbored complex and functionally important microbial communities. Yet much remains to be understood about how these coral-microbe symbioses evolved, and which key factors influence microbial communities in modern corals. Coral diversity is too great to individually assess the biotic and abiotic factors that maintain the microbiome of every coral species. The present challenge is thus to uncover general rules for the assembly of coral microbiomes that inform estimates of the effects of microorganisms in understudied portions of the coral tree. However, disentangling the many host and environmental features that influence the microbiome requires large and methodologically consistent surveys of phylogenetically diverse corals across geography

Many host-microbial symbiosis studies find correlations between host phylogenetic relationships and microbial community composition, a pattern known as phylosymbiosis[7]. Phylosymbiosis has been reported for the root microbiome of flowering plants;[8] the mesohyl of marine sponges;[9] insect microbiomes;[7,10] and the gut microbiome of terrestrial mammals[11] (including *Peromyscus* deer mice[7] and wild hominids[7]). Phylosymbiotic patterns can be explained by several mechanisms, including codiversification of abundant microbial lineages with their hosts, filtering of microbial communities by host traits, or coupling between host phylogeography and environmental effects on the microbiome[7,10,12]. We are only beginning to differentiate these alternatives[10], and studies accounting for the joint effects of phylogeny, geography, and host traits are sorely needed. Moreover, different animal secretions, tissues and organs typically harbor distinct microbiomes (e.g. ref. [13].) that may also show different patterns of phylosymbiosis, although this possibility has not yet been fully explored.

The phylum Cnidaria diverged prior to the bilaterian radiation. Thus, scleractinian coral microbiomes represent a key piece in the broader puzzle of how animal microbiomes arose. Coral mucus, tissue, and skeleton show distinct microbial community composition (e.g. refs. [14,15].), affording the opportunity to test whether they also show different patterns of phylosymbiosis. Additionally, the high diversity and wide geographic distribution of reef-building corals presents a natural experiment for testing how host traits and environmental context influence the microbiome, and are invaluable resources for understanding how modern host-microbial symbioses evolved.

Scleractinian corals have been diversifying for longer than some more commonly studied symbiotic systems such as flowering plants and placental mammals[16]. Their microbiomes are known to be partially species-specific (e.g., ref. [14].), and reports from other Cnidaria, such as gorgonians, suggest potential codiversification with *Endozoicomonas* symbionts[17,18]. Yet comparisons of six species of coral and an octocoral outgroup found microbiome similarities that seemed to better align with morphology than phylogeny[19], suggesting a strong influence of host traits on the microbiome. Whether scleractinian corals show phylosymbiosis in overall community composition or cophylogeny with specific bacteria or archaea has not yet been definitively established.

The abundance of overlapping factors that affect the coral microbiome is difficult to disentangle. Many host traits are highly correlated with one another due to phylogenetic constraints, and many environmental variables are correlated due to large-scale patterns of climate and geography. Thus, analyses of these variables cannot be conducted in isolation.

We designed a comprehensive sampling and analysis strategy that asked how the microbial communities residing in the mucus, tissue, and skeleton of diverse Australian corals were shaped by host phylogeny, host functional traits, geography and environmental variables. We collected DNA samples from the mucus, tissue, and skeleton of phylogenetically diverse scleractinian species, as well as selected outgroups and environmental references. We sequenced 691 16S rRNA gene libraries from these samples, primarily targeting bacterial and archaeal members of the microbiome. We paired these microbiome data with a multigene molecular phylogeny of scleractinian corals[20], coral functional traits from the Coral Trait Database[21], and extensive *in-situ* metadata (Methods)[22]. For questions that were sensitive to host phylogeny, we integrated these diverse datasets using phylogenetic Generalized Linear Mixed Models (pGLMMs). This approach provided a unified Bayesian framework in which to test hypotheses in coral-microbe coevolution and the influence of various environmental factors on coral-microbe symbiosis.

We show that coral microbiomes differ in richness, composition, and consistency across anatomy. In all anatomical compartments, both host and environment influence the microbiome. However, the relative influence of host vs. environmental parameters varies strongly across anatomy. We confirm phylosymbiosis in coral tissue and skeleton microbiomes, yet also present evidence that host-microbial cophylogeny influences microbial abundance for only a select subset of bacterial taxa associated with corals. Notably, that subset includes certain host-specific subclades of the prominent coral symbiont *Endozoicomonas*. Together, these results help to clarify how the evolution and ecology of the coral microbiome varies across anatomy.

## Results

**Data collection and workflow.** Coral, water, and sediment samples were collected from 21 sites around Australia spanning 17° of latitude. A total of 236 coral colonies were sampled from 32 scleractinian and 4 cnidarian outgroup taxa representing both the Hexacorallia and Octocorallia (Supplementary Data 1). Hexacorallia (anemones, corallimorpharians, zoanthids and scleractinian corals) and Octocorallia (gorgonians) are both monophyletic groups within class Anthozoa. A subset of corals was resampled at Lizard Island in summer and winter to assess seasonal effects. Up to 162 host and environmental metadata parameters were recorded or calculated for each sample (Supplementary Data 2). Combined, these data represent more than 425 million years of coral evolution[20].

A workflow summarizing the major analytical steps is presented in Supplementary Fig. 1. Coral samples were partitioned into mucus, tissue, and skeleton compartments (Methods), and sequenced alongside water and sediment samples from the same reefs, yielding a total of 691 samples for small subunit ribosomal RNA (16S rRNA) gene sequencing. These included 227 mucus samples, 223 tissue samples, 230 skeleton samples, and 11 additional reference samples (e.g., sediment and water; Supplementary Data S1). All samples were subjected to identical DNA extraction, PCR amplification using 515f/806r primers specific to the V4 region of the 16SrRNA gene of bacteria and archaea[23], and Illumina MiSeq sequencing. We note that despite the utility of 16S rRNA gene surveys, they are estimated to miss ~10% of environmental microbes[24], including certain archaea and the newly uncovered bacterial candidate phylum radiation[25].

Corals are regarded as challenging targets for DNA extraction. However, we found that the Earth Microbiome Project DNA extraction protocol provided sufficient DNA for analysis in most samples. After quality control, sequencing resulted in a total of 9,441,738 microbial reads (per sample median: 14,010; per sample mean: 13,664) partitioned across 129,305 unique OTUs (Methods, 97% similarity cutoff).

To avoid biases due to sequencing depth, we rarefied to even read depth (1000 sequences per sample) for most analyses (Supplementary Note 1). This strategy is conservative, in that it trades minimization of false positives for some loss of power. We also tested alternative rarefaction depths for characterization of core microbiomes (Supplementary Data 3), comparison of multivariate dissimilarities (Supplementary Data 4) and α-diversity across compartments (Supplementary Data 5). In the specific case of differential abundance testing, we either rarefied at 1000 reads/sample or used a parametric model without rarefaction (i.e., all pGLMMs, Methods) to maximize power from read depth in each sample. In total, we detected 69 microbial phyla associated with scleractinian corals (i.e., excluding outgroups), with 56.5% of the sequenced microbes in an average sample represented by Proteobacteria, while all Archaea represented just 2% of observed sequences (Supplementary Note 2).

To compare microbial community structure to host trees, we inferred a coral phylogeny using coral mitochondrial 12S rRNA gene sequences identified in our amplicon libraries (Methods), but constrained to match the topology of the multigene molecular phylogeny of corals published by Huang and Roy[20]. Typically, these unique mitochondrial sequences (mitotypes) had a resolution of around the coral genus level (but in some cases resolved species and intra-specific lineages), and were consistent with visual taxonomic identifications of the host. This had the effect of mapping this study's samples to the multigene Huang and Roy phylogeny wherever possible, while also estimating branch lengths among additional outgroup taxa and allowing for inclusion of samples that were not visually identifiable to the species level. A conceptually similar procedure (Methods) mapped microbial reads to the Greengenes 13_8 reference phylogeny[26].

**Drivers of coral microbiome structure vary across anatomy.** Coral tissue, mucus, and skeleton microbiomes differed in richness (Fig. 1a) and microbiome composition (Fig. 1b). Surprisingly, the coral endolithic skeleton was richer in microbial diversity than the tissue microbiome (Fig. 1a; Supplementary Note 3). Differences in microbiome composition between compartments were robust to choice of multivariate dissimilarity measures (Adonis permutational $p$ for Weighted UniFrac, Unweighted UniFrac and Bray-Curtis distance matrices<0.001; Supplementary Note 4).

Compartments differed in core microbiome membership (Supplementary Note 5), the fraction of the microbiome that was core (Supplementary Fig. 2a) and inter-colony variability (Supplementary Fig. 2b). Observed core microbiomes were consistent with past reports of the coral mucus core microbiome (Supplementary Note 6). Core microbiome analysis also confirmed the presence of Candidatus *Amoebophilus* (an intracellular symbiont of eukaryotes) as present in>50% of tissue microbiomes (consistent with;[14] see Supplementary Note 7). Overall, mucus microbiomes were notable for their relative stability between colonies and their high abundance of core vs. variable microbes (Supplementary Note 8).

Across all compartments, host species was the single most important variable structuring the coral microbiome in our data (Fig. 3, Supplementary Note 9, Supplementary Data 4). Broader

taxonomic levels were also associated with microbiome composition, with more specific taxonomic levels always explaining more microbiome variance than more general taxonomic levels. This finding held across several dissimilarity metrics and rarefaction depths, and in all 3 compartments (Supplementary Note 10, Supplementary Data 4). The influence of the coral host was thus a commonality of coral mucus, tissue, and skeleton microbiomes.

However, microbiomes associated with the three portions of coral anatomy differed in the extent of their relative responsiveness to host vs. environmental factors (Fig. 1c; Supplementary Fig. 3; Supplementary Data 6; additional discussion in Supplementary Note 11). For each environmental and host parameter, we tested its relative influence on coral mucus, tissue, and skeletal microbiomes (Fig. 1c). We then clustered host and environmental parameters in terms of their effects on the microbiome across compartments. Intrinsic host-based traits clustered separately from environmental traits. This was driven by the fact that environmental factors (e.g., season, temperature and turf algal competition) had a stronger influence on mucus microbiomes than tissue or skeleton microbiomes; whereas the coral species and its functional traits (e.g., growth form and disease susceptibility; Fig. 1c) had a stronger influence on tissue and skeletal microbiomes than mucus. Intriguingly, the diverse endolithic skeletal microbiomes were nearly as responsive to many host traits as the tissue microbiome (Fig. 1c), and showed the strongest response to the divide between Robust and Complex clade corals.

The finding that coral anatomical compartments differ in overall community composition raises the possibility that one could predict a given sample's compartment from knowledge of its microbial community. To quantify the accuracy with which a sample's compartment can be predicted from bacterial community membership at the genus level, a supervised classification model was developed using random forest analysis (a machine learning method). This model was 74.3% accurate—a 2.58-fold improvement on error rates compared to random guessing–demonstrating that a predictable set of bacteria are shared within compartments but differ among them.

We also used machine learning methods to quantify how much information the microbial community of each compartment conveyed about a suite of categorical host and environmental traits and host physiological and phylogenetic parameters (Supplementary Data 7). Consistent with our dissimilarity analysis (i.e., Fig. 1c), tissue microbiomes were better predictors of host factors like host genus (34% accuracy, 1.35x lower error rates than random guessing) and vertical transmission of *Symbiodinium* (86% prediction accuracy; 3.41-fold more accurate than random guessing) than were mucus microbiomes (Supplementary Data 7). Exploration of anatomical differences in coral microbiomes using machine-learning methods also revealed that the coral skeleton microbiome could better predict the deep phylogeny of the coral host (i.e., membership in the 'Complex' or 'Robust' clade) than could the microbiome of the coral tissue (Supplementary Results, Supplementary Data 7). Conversely, mucus communities were much better predictors of environmental features like contact with turf algae (82% accuracy; 2.14× lower error rates than random guessing) and sampling location (53% accuracy; 1.46× lower error rates than random guessing).

Together these multivariate and machine learning results clarify that while host species influences the microbiome across anatomy, the extent of host vs. environmental influence on coral microbiomes is not consistent in coral mucus, tissue, and skeleton. They further suggest that mucus microbiomes are useful for detecting environmental perturbations and that skeleton communities warrant greater attention as a diverse community strongly structured by host traits. Because coral compartments differed in both their composition and

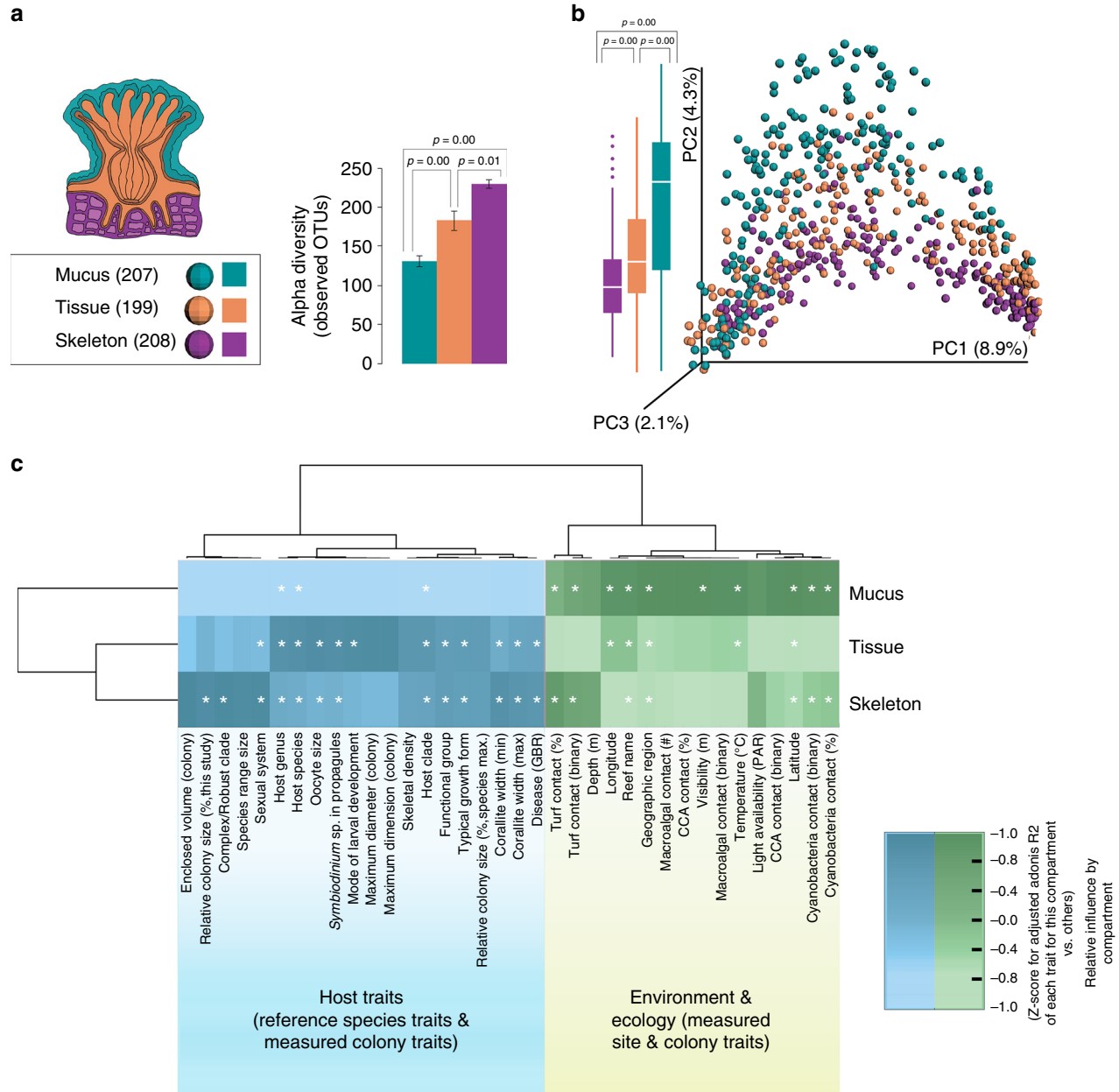

**Fig. 1** Anatomical differences in coral microbiomes. Coral mucus, tissue, and skeleton microbiomes differ in richness, composition, and response to host vs. environmental factors based on 16S rRNA gene sequence data. **a** Microbial community richness (observed OTUs) in coral mucus (teal), tissue (orange) and skeleton (purple), assessed at an even depth of 1000 reads per sample. $P$-values reflect Tukey's HSD. **b** Principal coordinates plot of coral-associated microbial communities (Unweighted UniFrac; $n = 614$). Reads were rarefied to 1000 reads per sample. Coral compartments show significant differences in community composition (Adonis $R^2 = 0.028$; permutational $p < 0.001$). The percent variation explained by the principal coordinates is indicated at the axes. Boxplots of the second PC elucidate differences among compartments. $P$-values reflect Tukey's HSD. **c** Relative influence of host and environmental factors on microbiome composition (Weighted UniFrac, Adonis adjusted $R^2$) in each compartment. Darker cells for a compartment indicate that it is more strongly influenced by that trait than the other compartments (Adonis adjusted $R^2$ values $z$-score normalized within columns). Cell values reflect adjusted $R^2$, which penalizes $R^2$ for each factor downward to allow for fair comparison among factors with varying degrees of freedom. Asterisks indicate a significant effect of that factor (Adonis permutational $p < 0.05$) on the microbiome in that compartment, following stringent Bonferroni correction across all traits and compartments. While both host and environmental factors influenced all compartments, host factors tended to influence coral tissue and skeleton more strongly than mucus, whereas host environment more influenced mucus microbiomes. All values in the table, plus other combinations of rarefaction depth and multivariate dissimilarity measure are presented in Supplementary Data 4

responsiveness to host and environmental variables, we report results of all subsequent analyses separately for each.

**Latitude and colony size influence the coral microbiome**. In addition to these general observations, two specific findings

emerged that bear mentioning (Fig. 2). First, the latitude of the sampling location significantly influenced the richness and composition of coral microbiomes. Moving away from the equator, coral microbiomes became less rich (Fig. 2a). This effect was significant for coral mucus and tissue, but not skeleton

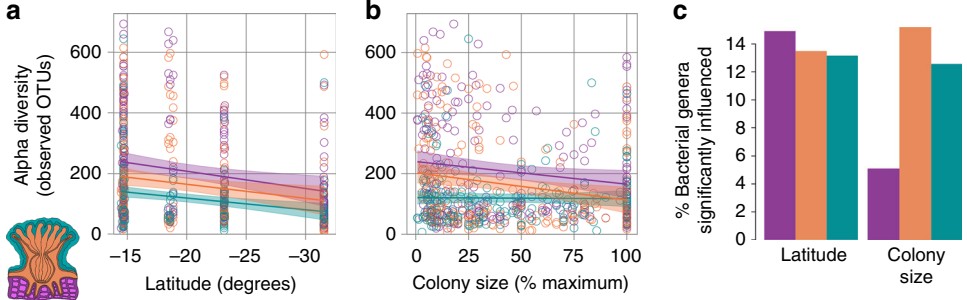

**Fig. 2** Effects of latitude and coral relative colony size on coral microbiomes. In all panels, we rely on phylogenetic Generalized Linear Mixed Models (pGLMMs; Methods), which account for potential confounding effects of coral phylogeny, for effect size and significance. **a** Microbial community richness (observed OTUs) as a function of latitude and coral anatomy (teal, coral mucus; orange, tissue; purple, skeleton). For visualization of latitudinal effects on richness, linear correlations are shown with colored lines, and their 95% confidence intervals are shown by shaded areas. Associations between latitude and microbiome richness were significant in coral mucus and tissue, but not skeleton (pMCMC: mucus, 0.0018; tissue, 0.0004; skeleton 0.468; pGLMM effect sizes: mucus, 0.026; tissue, 0.035; skeleton, 0.007). **b** Microbiome richness as a function of coral colony size relative to the maximum recorded size for each species and coral anatomy. Relative colony size vs. microbiome richness was visualized with linear regression. A negative association between coral relative size and microbiome richness was significant in tissue and skeleton, but a positive association in mucus was not significant (pMCMC: mucus, 0.86; tissue, 0.0008; skeleton, 0.02; pGLMM effect sizes: mucus, 0.028; tissue, −0.591; skeleton, −0.392). **c** Percent of tested microbial genera significantly associated with latitude and colony size in phylogenetically-controlled pGLMMs (Supplementary Data 6)

microbiomes, even after accounting for the uneven distribution of species across locations (Fig. 2a and legend). In addition to its effects on richness, latitude had a small but significant influence on microbiome composition, accounting for between 1–4% of variance in microbiome composition (Weighted UniFrac adjusted Adonis R$^2$), depending on the compartment (Fig. 1c). Phylogenetic GLMMs were fit separately to each compartment and showed that more bacterial genera were positively than negatively correlated with latitude (11, 13, and 9% of genera positively correlated with latitude in mucus, tissue, and skeleton, compared to 4, 1, and 4% negatively correlated with latitude in these compartments) (Fig. 2; Supplementary Data 6). Together these results suggest that patterns of diversity in coral microbiomes may mirror latitudinal diversity gradients seen in free-living communities[27].

We also found that proportionally larger corals (those closer to their species' maximum recorded size) showed differential microbiome composition and richness relative to smaller specimens (Fig. 2b). Effects of coral relative size on microbiome richness were significant in coral tissue and skeleton, but not mucus, after accounting for phylogeny (Fig. 2b). Coral size also had minor but statistically significant effects on microbiome composition in coral skeleton (Supplementary Note 12; Supplementary Data 4). Two bacterial genera, *Aurantimonas* and *Balneola*, were significantly reduced in larger corals (Supplementary Data 6). *Aurantimonas* has been proposed as the causative agent of White Plague Type II[28] and *Balneola* was previously identified as an indicator of sewage pollution in the Red Sea[29]. These trends may reflect increased vulnerability of smaller corals to opportunistic pathogens or may simply reflect normal shifts in microbiome composition over the course of coral development[30].

**Coral phylogeny influences microbiome diversity**. Phylosymbiosis refers to the evolutionary pattern in which the phylogeny of a related group of host organisms correlates with changes in multivariate community dissimilarities among their microbiomes[31]. Mantel tests assessed phylosymbiosis in the coral microbiome (Supplementary Data 8). These tests quantify the correlation between matrices of coral host phylogenetic distances and multivariate dissimilarity as measured by the Bray-Curtis or Weighted UniFrac measures. Using Bray-Curtis dissimilarities, more closely related corals had more similar microbiomes in both

tissue (Mantel $r = 0.16$, $p = 0.0001$; Supplementary Fig. 4a, dashed red regression line) and skeleton compartments (Mantel $r = 0.18$, $p = 0.0001$), but not mucus (Mantel $r = 0.02$, $p = 0.18$). Using the phylogenetically-aware Weighted UniFrac method deemphasized fine variation at the tips of the microbial tree, resulting in a significant signal of phylosymbiosis in skeleton, but not tissue or mucus microbiomes (Supplementary Data 8).

Our Mantel test results demonstrate patterns consistent with phylosymbiosis in skeleton and perhaps tissue microbiomes, but do not clarify the evolutionary scales over which these patterns emerged. Therefore, we used Mantel correlograms to assess how these correlations varied across multiple scales of phylogenetic divergence. Across all compartments and dissimilarity measures, microbiomes of the most closely related coral hosts were significantly more similar than expected by chance (Mantel $r > 0$; $p < 0.05$). In general, tissue and skeletal microbiomes became gradually more dissimilar throughout the entire range of host phylogenetic distances (Supplementary Fig. 4b, Supplementary Data 8). Mucus microbiomes, on the other hand, did not become more dissimilar as host phylogenetic distance increased past the second distance class (Supplementary Data 8).

We employed a similar procedure to test the evolution of microbiome richness in corals. As richness is a univariate rather than multivariate quantity, we conducted these tests using Moran's $I$ as a measure of univariate autocorrelation. These phylogenetic correlograms demonstrated that, like community composition, richness was significantly more similar among closely related corals than expected by chance (Moran's $I$ 95% lower CI > 0; Supplementary Fig. 4c, red confidence intervals). Additionally, richness was more *dissimilar* than expected among samples that were separated by phylogenetic distances of approximately 0.2 to 0.25 (Moran's $I$ 95% CI < 0; Supplementary Fig. 4c; blue confidence intervals), which corresponded roughly to between-family distances in our tree. At greater phylogenetic distances, they were no more or less similar than expected. These trends were consistent across coral mucus, tissue, and skeleton microbiomes. Coral microbiome richness is therefore influenced by the evolutionary histories of host corals. Importantly, the scales of phylogenetic divergence at which these effects appear, suggest that the radiation of modern reef-building coral families (between roughly 25 and 65 mya) was accompanied by large changes in microbiome richness, with changes continuing to accumulate during more recent speciation events. What's more,

these results demonstrate that the phylogenetic histories of corals partially constrain the composition of their tissue and skeletal microbiomes and the richness of all coral compartments. In other words, corals and their microbiomes exhibit phylosymbiosis[32].

**Limited cophylogeny despite phylosymbiosis.** Phylosymbiosis results from a number of different mechanisms: the steady evolution of host traits that directly influence the microbiome (e.g., by excluding certain microbes); spatial patterning of hosts that indirectly influence the microbiome via environmental or ecological interactions (e.g., dispersal to areas with intensive turf algae competition), or long-term codiversification between hosts and specific microbial symbionts[12,33].

To address these alternatives, we tested all microbial genera for associations with tips of the coral tree (host identity) or wider regions of the coral tree (host phylogeny). Genera were defined based on Greengenes taxonomic annotations, and therefore included some imprecise pools of unannotated taxa, but were deemed sufficient for our intended analyses (a complementary fine-scale approach is pursued below). Both host identity and host phylogeny were assessed using the genus-resolution 12S mitochondrial RNA gene markers extracted from amplicon libraries for each sample, and pGLMMs (see Methods) were used to separate the effects of environmental and physiological variables from host effects.

Even after accounting for some of the most important environmental and physiological factors from the multivariate analysis (e.g., geographic region, turf algal contact, disease susceptibility, and maximum corallite width), most microbial genera showed host-specific abundance patterns (Supplementary Table 5). Yet most coral-associated microbes were correlated with host identity, not host phylogeny. Phylogenetic GLMMs estimated that the abundances of between 62% (tissue) and 75% (mucus) of microbial genera were significantly correlated with at least one host mitotype (host identity), depending on the compartment analyzed (Supplementary Data 6). For example, 100/446 microbial genera detected in tissue microbiomes were significantly more abundant in the *Acropora* mitotype than in others. Overall, 276/446 (62%) of microbial genera were associated with a host mitotype, but only 13/446 (3%) of genera in coral tissue were associated with host phylogeny (Supplementary Data 6). Genera associated with host phylogeny include *Candidatus* Amoebophilus (Cytophagales: SGUS912), a taxon previously identified as a core coral microbiome member across three species[15]. *Candidatus* Amoebophilus was associated in the skeleton with the coral clade formed by both *Seriatopora* and *Stylophora*, rather than with individual 'host identity' mitotypes (for additional discussion see Supplementary Results). Mucus microbiomes showed fewer genera (1.6%) associated with host phylogeny than tissue, while skeletal microbiomes showed more (4.9%). Taken together, this analysis confirmed that while the coral microbiome is highly host-specific, only a restricted subset of the microbiome members show preferences for entire groups of related corals.

**Cophylogeny identifies 4 long-term coral-bacterial symbioses.** The above GLMM analyses were conducted at the level of microbial genera, but finer-scale taxonomic variation is likely to exist. Also, the previous analysis identified only the response of microbial genera to host phylogeny, rather than any potential interactions between microbial and coral phylogenies. Therefore, we ran pGLMMs incorporating both coral and microbial phylogeny on fine-scale microbial sequence variants using Minimum Entropy Decomposition (MED)[34]. These methods tested whether corals showed patterns of cophylogeny with any of their microbial

associates, which in this context refers to the tendency for groups of related microbes to be associated with groups of related hosts. Such patterns can arise from coevolution or codiversification, and may thus be a sign of intimate symbiosis, mutualistic or otherwise.

Because these analyses are computationally intensive, only the most prevalent microbial taxa were tested. A total of 25 bacterial family-level groups present in>50% of samples from at least one coral compartment were selected for detailed analysis. All but two of the families tested had some form of host specificity, with members either associated with particular coral mitotypes (representing species or genera) or particular regions of the coral tree. More formally, each of these bacterial families showed one of the following interaction effects (Fig. 3): host identity by bacterial identity; host identity by bacterial phylogeny; host phylogeny by bacterial identity; or host phylogeny by bacterial phylogeny (i.e., cophylogeny). One-to-one associations between individual bacterial sequence variants and individual coral hosts (i.e., host identity by bacterial identity interaction effects) were only significant in three bacterial groups: Clostridiaceae (mucus), unclassified Myxococcales (mucus), and unclassified Kiloniellales (mucus and tissue). In 19 of 25 families, host identity interacted significantly with bacterial phylogeny, meaning individual coral mitotypes were significantly associated with clades of related bacteria. However, the converse pattern did not occur: no individual bacterial sequence variants were significantly associated with clades of related coral hosts.

Four bacterial groups exhibited significant cophylogenetic effects (i.e., host phylogeny by bacterial phylogeny interaction): Clostridiaceae, *Endozoicomonas*-like bacteria (Endozoicomonaceae in Greengenes), unclassified Kiloniellales, and unclassified Myxococcales (Fig. 3, red box). Cophylogeny in *Endozoicomonas*-like bacteria was detected in both tissue and mucus (ICCs, 95% lower bounds: 0.20 and 0.17, respectively) (Figs. 3, 4). Cophylogeny in Clostridiaceae and unclassified Myxococcales was detected within the coral skeleton only (ICCs, 95% lower bounds: 0.06, and 0.29, respectively), and it was detected in unclassified Kiloniellales in only the tissue compartment (95% ICC lower bound: 0.03).

Together, these results show that while overall coral microbiome composition and richness do track phylogeny, and the majority of microbial genera show significant host-specificity, only a small subset of coral-associated microbial diversity shows larger-scale interactions between coral and bacterial phylogeny.

***Endozoicomonas* clades vary in host preference and host range.** *Endozoicomonas*-like bacteria are important coral symbionts[35], and in our data showed the strongest signal of cophylogeny among bacteria found in coral tissues. We therefore analyzed this group in greater depth. Inspection of the phylogeny of *Endozoicomonas*-like bacteria (Methods) revealed two major coral-associated divisions within the group (Fig. 4, Supplementary Fig. 4): one in which most strains were host-specific (hereafter 'Clade HS' for 'Host-Specific'), and another where most strains had a cosmopolitan distribution across multiple hosts (hereafter, 'Clade HG' for 'Host Generalist).

Within the host-specific clade HS, two bacterial sub-clades were strongly associated with the two major lineages of corals ('Complex' or 'Robust' corals). We have termed these clades of *Endozoicomonas*-like bacteria 'HS-R' for 'Host-Specific: Robust' and 'HS-C' for 'Host-Specific: Complex'. All of these clades and subclades were well-supported by posterior probabilities (posterior probabilities: Clade HG, 1.00; HS-R 0.92; HS-C 0.72) with the

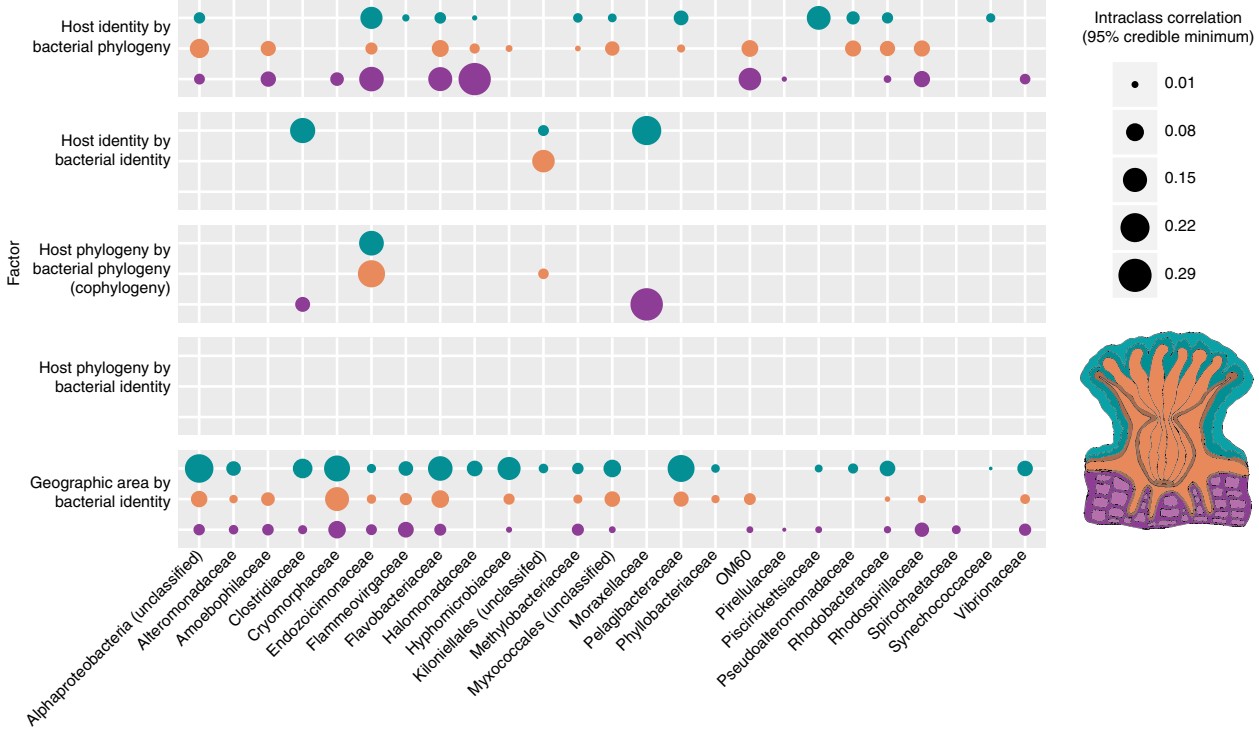

**Fig. 3** Effects of host identity, phylogeny, and cophylogeny on bacterial families. Results are derived from co-phylogenetic GLMM analysis within prevalent bacterial families, and incorporate geographic area, bacterial and coral host identity, and bacterial and coral host phylogeny (see Methods and workflow in Fig. 1). Each block of rows corresponds to a factor in the model (main effects are not shown). Each row within a block corresponds to a tissue compartment (teal = mucus, orange = tissue, and purple = skeleton; see coral polyp illustration), while each column corresponds to an independent model fit for the specified microbial group. Dots were plotted only for 'significant' factors (ICC lower bound > 0.01). The size of each dot represents the intra-class correlation coefficient (ICC) 95% credible lower bounds from co-phylogenetic linear model analysis. While coral host identity is associated with bacterial phylogeny for most prevalent bacterial families (top block of rows), only 4 bacterial families show co-phylogeny with corals (middle block of rows)

exception of Clade HS, which was only weakly supported (posterior probability 0.33).

To further assess the relationship between corals and Clade HS *Endozoicomonas*-like bacteria, we fitted a GLMM that included all corals but only clade HS bacteria, and another that included only the Robust clade corals and clade HS-R bacteria. The cophylogeny terms from both these tests were highly significant (ICCs, 95% lower bounds: 0.34 and 0.21, respectively).

The coral-associated members of Clade HS from this study were more closely related to *Endozoicomonas* strains previously reported to live in symbiosis with diverse non-scleractinian hosts (gorgonians, mollusks, sponges, and other marine invertebrates) than they were to members of Clade HG. Thus it appears that *Endozoicomonas*-like bacteria have formed novel associations with scleractinian corals multiple times throughout their evolution, but that such host-swapping has been a relatively rare occurrence.

## Discussion

Scleractinia have been diversifying for almost half a billion years[20]. In about half that timespan, flowering plants appeared, diversified and evolved important specialized microbial symbioses in specific lineages[36]. Here we demonstrate that a phylogenetic framework for analysis of coral microbes can reveal how scleractinian corals' evolutionary history, host traits and the local environment interact to shape coral microbiomes. Our results test longstanding hypotheses that bear on potential coral-microbe coevolution, and add quantitative details and

taxonomic breadth to several previously explored patterns in coral microbiology.

We originally hypothesized that corals would show signs of phylosymbiosis throughout their entire phylogenetic history. While our results are in accord with this hypothesis in coral skeleton and tissue, the same is not true for the coral mucus microbiome. Despite documented variability in the chemical composition of coral mucus between species[32], and significant host-specificity in the mucus microbiome, host specificity in the mucus microbiome was limited to relatively recent divergences and was not significantly structured by larger scales of host phylogeny. Importantly, because this analysis focused on the entire Scleractinian order, it did not test whether patterns of phylosymbiosis occur in the mucus within specific coral lineages or at intrageneric timescales generally. In contrast to the patterns in the mucus, the coral skeleton, which has been less intensively studied than mucus and tissue, showed both the greatest microbiome richness and the strongest signal of long-term phylosymbiosis. These findings emphasize that different anatomical regions of animal hosts may show distinct evolutionary patterns. This observation will be relevant for studies in other systems (e.g., mammals) where most studies of host-microbe coevolution have emphasized a single body site (e.g., the distal gut).

Phylosymbiosis can emerge as a consequence of multiple mechanisms, including codiversification of many lineages, microbial habitat filtering by host traits, or the interaction of host and microbial biogeography[12]. We tested whether the most prevalent coral-associated bacteria demonstrated cophylogenetic

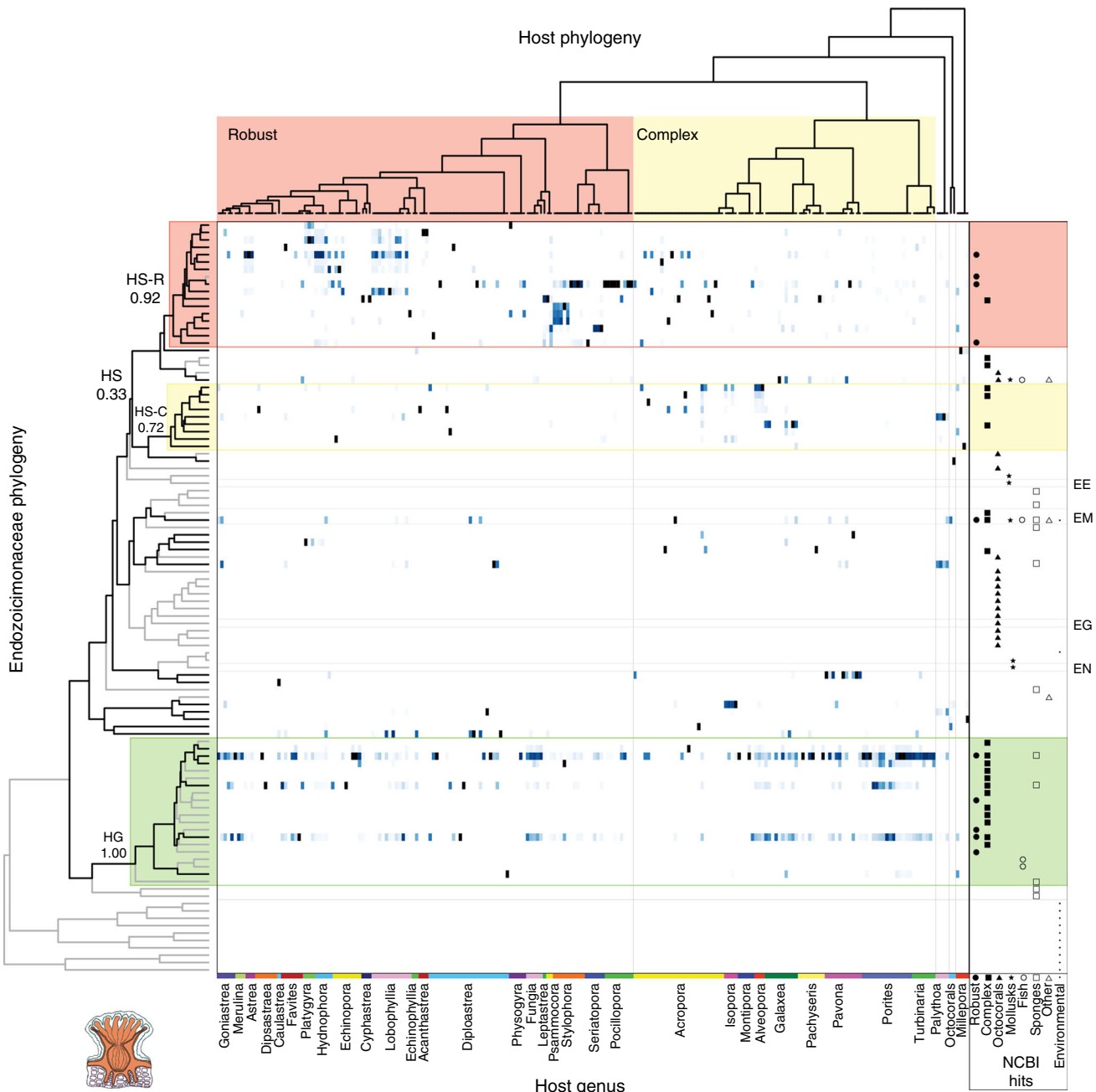

**Fig. 4** Distribution of *Endozoicomonas*-like bacteria across coral hosts. The heatmap illustrates patterns of association between *Endozoicomonas*-like bacteria and coral hosts. Colored cells represent the relative abundance of *Endozoicomonas*-like bacterial sequences (out of the total abundance of *Endozoicomonas*-like bacteria) in each coral host, plotted on a scale from 0% (white) to 100% (dark blue). The x-axis is arranged by coral host phylogeny, which is shown at the top. The y-axis is arranged by the phylogeny of the most abundant *Endozoicomonas*-like bacterial sequences observed in host tissues, which is shown to the left (Bayesian posterior support values are shown for clades of interest). Clade HG (Host Generalist; green box) is prevalent in diverse species spanning both the Complex and Robust clades. Clades HS-R (Host-specific: Robust; pink box) and HS-C (Host-specific: Complex; yellow box) are composed of host-specific *Endozoicomonas*-like lineages. On the right, the host organisms of each sequence variant's perfect matches in NCBI's nr database are shown. Cultured and named strains are identified with abbreviations (EE: *Endozoicomonas elysicola*, EM: *Endozoicomonas montiporae*, EG: *Endozoicomonas gorgoniicola*, EN: *Endonucleobacter bathymodioli*). Sequences in clades HS-R and HS-C are consistently associated with Robust and Complex clade corals, respectively (see Supplementary Fig. 5 for more detail)

patterns with their hosts. We used pGLMMs to compare the prevalence of individual sequence variants within particular bacterial families among diverse coral hosts. This approach allowed us to disentangle the effects of geographic area, cophylogeny, and distinct associations between individual hosts and microbes. Of the 25 bacterial families tested, cophylogentic

interactions significantly influenced the abundance of only 4. Thus, although many coral-associated bacteria are host-specific, and the overall composition of coral microbiomes tracks phylogeny (i.e., phylosymbiosis), only a select minority of coral-associated bacterial families show cophylogenetic signals consistent with long-term host-microbe codiversification. This result

emphasizes that while host-microbe cophylogeny likely contributes to phylosymbiosis, other factors, such as biogeographic effects and phylogenetically patterned host traits are likely very important in producing this pattern.

That coral microbes differ in their extent of cophylogeny with their host also emphasizes that the microbiome is not a single unit of selection, but instead contains diverse players that vary greatly in the extent of their history of association with the host and one another[12]. Host specificity of certain microbes with extant coral species, community-level phylosymbiosis across the coral microbiome, and cophylogeny of certain microbial lineages with their host are all distinct concepts that should be distinguished. We recommend that observations of phylosymbiosis be accompanied by finer-scale tests of host-microbe cophylogeny in order to identify specific microbial lineages that may have coevolved or codiversified with their hosts. These may warrant additional investigation as potential 'key players' in the microbiome, but a time-calibrated microbial phylogeny would be necessary to test for it.

The microbial families that show signs of cophylogeny may be associated with important host functions that have led to stable symbiotic relationships across extended evolutionary time. The pGLMM methods used here allow identification of such taxa from a broader symbiotic community, even in the absence of strict one-to-one associations between hosts and symbionts, and will be relevant to other study systems. We identified four bacterial lineages displaying signs of cophylogeny with their coral hosts, all of which represent important targets for future study. One lineage, the *Endozoicomonas*-like bacteria, has previously been hypothesized to have codiversified with their coral hosts throughout the evolution of Scleractinia[37]. These results are relatively consistent with this hypothesis for one subclade of *Endozoicomonas*, but suggest that the abundant variants found in well-studied *Porites* corals are more cosmopolitan in their distribution. A greater geographic breadth of samples and representatives of the many azooxanthellate scleractinians will help inform this notion further, and a test of codiversification specifically will require a better-resolved and time-calibrated *Endozoicomonas* phylogeny.

In addition to the *Endozoicomonas*, three other groups of functionally distinct bacteria showed significant patterns of cophylogeny: the proposed mutualist Kilioniellales[38], the predatory 'wolf pack' bacteria Myxococcales, and a group of organisms generally hypothesized to be pathogens of corals, the Clostridiaceae[4,38–40]. Future work on these taxa may provide insight into their broader roles in coral evolution and health.

Our survey of Australian coral microbial diversity provides the most conclusive evidence to date that phylosymbiosis has occurred between corals and their microbiomes. Despite this, cophylogeny between scleractinian corals and their microbial symbionts is likely restricted to a small subset of bacterial families. The results of this survey further quantify the relative influence of host and environmental drivers on the microbial diversity of coral mucus, tissue and skeleton. A still more comprehensive picture of coral microbiology will be gained with future efforts that expand analyses to global sample datasets (including potentially informative samples from deep-water or Caribbean corals), development of improved statistical models (e.g., by relying less on arbitrary taxonomic thresholds; see for instance the emerging 'ClaaTU' method[41]), and connection of these patterns of microbial diversity to other members of the coral microbiome such as *Symbiodinium*. In particular, the addition of deep-water, azooxanthellate corals could fill in important gaps in the phylogeny and help test the generality of phylosymbiosis in coral microbiomes.

## Methods

**Selection of target sites**. We aimed to collect coral specimens spanning coral phylogenetic diversity from a variety of Australian reefs. We targeted collection based on the 21 major coral clades defined in one of the most recent molecular phylogenies available at the start of the project[33]. Many of these monophyletic groups have since been defined as family-level taxa. Corals were collected at several sites on the east and west coast of Australia. These included Ningaloo Reef (Western Australia), Lizard Island, multiple reefs along the northern Great Barrier Reef, and Lorde Howe Island. Samples at Lizard Island were collected in both Summer and Winter, allowing for comparison of seasonal effects at one site across diverse corals.

**Collection of metadata**. During sampling, each coral, outgroup species, water, and sediment sample was associated with MIxS metadata[42]. This was accomplished by recording standardized metadata about each site prior to dives, and using an underwater metadata sheet (available at https://doi.org/10.6084/m9.figshare.5326870.v1). These metadata included basic features of coral species (as identified in the field), location, depth, water temperature, but also diver annotation of contact with macroalgae, turf algae or cyanobacteria (and the percent of the coral in contact); the presence of any visible tissue loss or disease signs; and coral color (using the Coral Reef Watch color charts[43]. Additionally, photographs of each coral were taken and released via openly accessible third party websites. They are easy to browse and thoroughly keyworded with taxonomy, location, and sample ID metadata on Flickr: https://flic.kr/s/aHsk9mjb54, and permanently archived in raw camera format with a spreadsheet linking filenames to colony names on FigShare at https://doi.org/10.6084/m9.figshare.5318236.v2.

**Coral sampling**. All coral samples were collected by AAUS-certified scientific divers, in accordance with local regulations. Relevant permit numbers are: CITES (PWS2014-AU-002155, 12US784243/9), Great Barrier Reef Marine Park Authority (G12/35236.1, G14/36788.1), Lord Howe Island Marine Park (LHIMP/R/2015/005), New South Wales Department of Primary Industries (P15/0072–1.0, OUT 15/11450), US Fish and Wildlife Service (2015LA1632527, 2015LA1703560), and Western Australia Department of Parks and Wildlife (SF010348, CE004874, ES002315). Only healthy corals were collected.

One goal of the project was to compare microbial diversity associated with the coral mucus, tissues and skeletons across many coral colonies. Each of these compartments represents a simplification of more complex structure, and much work remains to be done on the finer-scale distribution and dynamics of microorganisms across coral anatomy. For this project, we felt that a consistent reporting of these compartments across diverse corals represented a tractable step forward, given the scale of the project. Mucus was collected by gently agitating the surface of corals for ~30 s with a blunt 10 mL syringe. Exuded mucus or surface water (if no visible mucus was exuded) was then collected by suction. On the surface, settled mucus typically formed a distinct visible layer within the syringe. This was expelled into a cryogenic vial and stored in a dry shipper charged with liquid nitrogen for subsequent processing.

Tissue and skeletal samples were collected from each colony by hammer and chisel, or (for branching corals) by bone shears. These fragments were placed in sterile WhirlPaks and returned to the surface where they were snap frozen in a liquid nitrogen dry shipper until processing. In the laboratory, tissue was washed with sterile seawater (which removed visible mucus and detritus), then separated from skeleton using pressurized air of between approximately 800 and 2000 PSI (an 'air gun'). Skeleton was sampled using a sterile chisel to isolate a ~1 cm³ region of skeleton that was not in direct contact with coral tissue. Skeleton samples were collected without regard to endolithic algae presence or absence (i.e., endolithic algae were neither specifically targeted nor excluded). Tissue slurries and skeleton samples were added directly to a MoBio PowerSoil Kit (MoBio Laboratories, Carlsbad, California) bead tube (which contains, among other things, a solution of guanidinium preservative) and stored at −80 °C until DNA extraction.

**Sampling of reference samples**. Because reef water and adjacent sediment might have an effect on the microbiota of corals from the same reef (especially in coral mucus), reef water and sediment were sampled at multiple sites. Surface seawater samples (1 L) were filtered through 0.22 μm Millipore Sterivex filters (Sigma-Aldrich, St. Louis, MO, USA) and reef sediment samples (2 mL) were collected in sterile cryogenic vials. Samples were snap frozen in a liquid nitrogen dry shipper, and subsequently stored at −80 °C until DNA extraction.

For comparison with corals from the same reef, we also opportunistically sampled non-scleractinian cnidarians from the genera *Millepora* (fire corals), *Palythoa* (zoanthids), *Heliopora* (blue corals), and *Lobophytum* (soft corals).

**16S library preparation, sequencing, and initial quality control**. DNA was extracted from skeleton, tissue, mucus, and environmental samples using the MoBio Powersoil DNA Isolation Kit. Two-stage amplicon PCR was performed on the V4 region of the 16S rRNA gene using the 515 F/806 R primer pair that targets bacterial and archaeal communities[23]. Extraction blank controls were also included in amplification and sequencing for quality assurance. The average concentration

of extracted DNA used for PCR was 10.8 with a standard error of 1.2. First, 30 PCR cycles were performed using 515 F and 806 R primers (underlined) with linker sequences at the 5′ ends: 515F_link (5′-ACA CTG ACG ACA TGG TTC TAC AGT GCC AGC MGC CGC GGT AA−3′) and 806R_link (5′-TAC GGT AGC AGA GAC TTG GTC TGG ACT ACH VGG GTW TCT AAT−3′). Each 20 µL PCR reaction was prepared with 9 µL 5Prime HotMaster Mix (VWR International), 1 µL forward primer (10 µM), 1 µL reverse primer (10 µM), 1 µL template DNA, and 8 µL PCR-grade water. PCR amplifications consisted of a 3 min denaturation at 94 °C; 30 cycles of 45 s at 94 °C, 60 s at 50 °C and 90 s at 72 °C; and 10 min at 72 °C. Next, amplicons were barcoded with Fluidigm barcoded Illumina primers (8 cycles) and pooled in equal concentrations for sequencing. The amplicon pool was purified with AMPure XP beads and sequenced on the Illumina MiSeq sequencing platform (using V3 chemistry) at the DNA Services Facility at the University of Illinois at Chicago.

QIIME (v1.9)[44] was used to process all 16S sequence libraries. Primer sequences were trimmed, paired-end reads merged, and QIIME's default quality-control parameters used when splitting libraries. Chimeras were removed and 97%-similarity OTUs picked using USEARCH 7.0[45], QIIME's subsampled open-reference OTU-picking protocol[46], and the 97% Greengenes 13_8 reference database[26]. Taxonomy was assigned using UCLUST, and reads were aligned against the Greengenes database using PyNAST[47]. FastTreeMP[48] was used to create a bacterial phylogeny with constraints defined by the Greengenes reference phylogeny. Following quality control, 9,441,738 usable reads remained. The number of per sample reads ranged from 2 to 38,523 with a median of 14,010, mean of 13,644, and standard deviation of 7565. Reads were partitioned across 129,305 unique OTUs (97% similarity cutoff). Sequencing success did not show any obvious trends with regards to host taxonomy or geographic location.

A 'canonical' rarefied OTU table was created and used for all downstream analyses except the linear model analyses. To create this table, OTUs were filtered out of the starting table if their representative sequences failed to align with PyNAST to the Greengenes database or if they were annotated as mitochondrial or chloroplast sequences. The beta_diversity_through_plots.py script was then used to rarefy the resulting table to exactly 1000 sequences per sample, and to calculate from this rarefied table multivariate dissimilarity measures including Bray-Curtis, Binary Jaccard, Weighted UniFrac, and Unweighted UniFrac. Also from this table, α-diversity statistics were calculated using alpha_rarefaction.py, including the number of OTUs observed, evenness, and Faith's Phylogenetic Diversity.

**Mitochondrial annotation and quality control**. The primers used in this study were designed to selectively amplify the V4 region of bacterial and archaeal 16S rRNA gene, but we have noticed in many of our studies that they (and other standard primer sets) tend to strongly amplify corals' mitochondrial 12S rRNA gene, which is the homolog of the bacterial 16S rRNA gene. Because our samples included species that were not in the Huang and Roy 2015[20] phylogeny (including, critically, all outgroup taxa), these 'off-target' host mitochondrial reads were used to inform phylogenetic analyses and for an additional layer of quality-control. First, split_libraries_fastq.py was run on the raw forward reads without any quality trimming. Then, primers and adaptor sequences were removed, and USEARCH used to de-replicate 100% identical sequences. A frequency table was created and the data were filtered to contain only sequence variants with a total count of at least 100. Greengenes taxonomy was assigned to the remaining sequence variants with UCLUST as before, and sequence variants that had no match in the Greengenes database (e.g., putative non-bacterial or archaeal sequences) were isolated. For each host species, sequence variants were manually submitted to NCBI's BLASTn web interface in order of their total abundance, comparing against the entire nr database. If a variant's top 20 hits were annotated as coral mitochondria of any species, the sequence was copied to a FASTA file of host sequences. If all three compartments of a single coral individual of the same putative species still had unclassified variants that were more abundant, then manual annotation of those variants continued until either another coral mitochondrial sequence variant was found or there were no more variants in those samples that were more abundant than the previously annotated mitochondria. Using this method, no host sequences were found for some species of coral. The process was repeated for these species individually, without first discarding sequences that had counts of less than 100. In this way, mitochondrial sequences were eventually identified for every sample in the study.

Once all host species mitochondrial sequences were identified, the original frequency table of all unique sequence variants was filtered to contain only the identified host sequences. For each individual sample, the most abundant mitochondrial type was determined, and this information was then added to the sample's metadata as its '12S genotype'. Then, all selected host sequences were aligned using MAFFT[49] and de novo phylogenies were constructed in BEAST 2.4.2[50], with a chain length of 10 million, thinning interval of 1000, a log-normal relaxed clock model, and the site model selected using bModelTest[51]. The maximum clade credibility tree was selected using TreeAnnotator with a burn-in of 25% and common ancestor heights. This tree was compared to the expected topology (monophyletic Anthozoa, Hexacorallia, and Scleractinia, and otherwise matching the Huang and Roy 2015[20] molecular tree) to identify potential mismatches among the observed sequences and the field species identification.

Regions of the tree with topology that differed from expectation were manually inspected.

Using this strategy, two coral individuals were noted whose field identifications placed them in the family Merulinidae, but whose sequence variants were strongly indicative of a relationship with the family Lobophylliidae. In these instances, further analyses verified that the same mitotype was detected in all three compartments of the same individual. Photos from collections in the field were consulted, and both were ultimately determined to have been misidentified in the field and in fact belonged to the genus *Echinophyllia*. Their metadata and annotations were updated to reflect this.

Aside from these two taxa, it was determined that unexpected topology in the de novo phylogeny was a result of imprecise resolution of the 12S V4 marker. For example, sequences from *Millepora*, *Palythoa*, and both octocoral species were placed in a monophyletic clade including the Complex corals, though they properly belong as outgroups to all Scleractinia. These errors emphasized the limitations of our opportunistic host sequence data to build a de novo phylogeny. Thus, having confirmed identifications of host species to within the resolution of the 12S marker, a new phylogeny was constructed with the topology constrained to exactly match the Huang and Roy 2015[20] molecular phylogeny and the known relationships of outgroup taxa. In cases where a single 12S genotype belonged to members of a polyphyletic group of taxa, we created separate tips for each monophyletic group. The mitochondrial sequence alignment and BEAST 2.4.2 were used to estimate relative branch lengths on this tree by supplying the starting tree and turning off all topology operators. The resulting tree was used for all phylogenetic analyses. As the branch lengths in this tree are derived from a relaxed clock model and limited sequence data, they are likely to represent some average between divergence times and degree of molecular evolution. Thus, analysis using these branch lengths represents a compromise between assuming correlation of traits is proportional to time since divergence and assuming that correlation of traits is proportional to overall evolutionary change since divergence.

**Annotation of coral life history strategy**. To assess connections between coral traits and microbiome structure, coral species sampled in this study were mapped to functional features. These host features were added to the microbial mapping file, and used for tests of microbiome structure vs. host traits.

Coral life history strategies from Darling et al[52]. ('weedy', 'competitive', 'stress-tolerant', and 'generalist') were digitized and associated with coral species. Some species have recently been moved between genera based on updated phylogenetic evidence[53]. In these cases, both the original species name and the revised name are noted in the metadata. In some cases, species sampled were not annotated in Darling et al[52]. These were not assigned an annotation if annotated members of the same genus had mixed life-histories, or if only a single species of the same genus had been annotated. In cases where at least two members of the genus had been annotated and all annotated members shared the same life-history strategy, the same annotation was assigned to other members sampled from the genus.

**Annotation of coral functional traits**. Metadata associated with each species sampled was annotated with 28 reproductive, biogeographic, and morphological traits from the Coral Trait Database (CTDB) v. 1.1.123. These traits included basic details on coral distribution (abundance worldwide and on the Great Barrier Reef, range size, northerly and southerly limits, upper and lower depth limits), reproduction (sexual system, mode of larval development, propagule size, presence of Symbiodinium in propagules), phylogeny (genus and species ages), morphology (growth form, skeletal density, corallite maximum width, maximum growth rate) and conservation (IUCN Red List Category).

**Adonis analysis of factors affecting microbial composition**. We tested the influence of multiple host and environmental factors on the microbial community of each compartment. These results are presented in Fig. 1c and Supplementary Fig. 2, while the raw underlying data is presented in Supplementary Data S3. Throughout the analysis care was taken to account for the effects of rarefaction depth (we tested the robustness of the results at rarefaction depths of 1000 or 10,000 sequences/sample), β-diversity distance measure (we tested three distance measures), the degrees of freedom in each parameter (we used adjusted $R^2$ values to account for differences in degrees of freedom), and to stringently control for the number of comparisons performed (using Bonferroni correction).

β-diversity distance matrices were calculated from separate OTU tables for coral mucus, tissue, and skeleton (outgroups and environmental samples were not included in this analysis). We calculated distance matrices using Weighted UniFrac distances, Unweighted UniFrac distances or Bray-Curtis dissimilarities. Then, for each host or environmental factor, the distance matrix was filtered to just those samples for which metadata were available (i.e., excluding 'Unknown' values). This prevented 'Unknown' values from being treated as a bona fide category in downstream statistical tests. The filtered distance matrix was then tested for clustering by factor using permutational tests (as implemented in Adonis in QIIME 1.9.1; 999 permutations per test). Because categories that can take on more values (e.g., species) are biased upwards in raw $R^2$ values, we calculated adjusted $R^2$ values for each category. These adjusted $R^2$ values are primarily useful in that they allow for fair comparison between factors with differing degrees of freedom. Therefore,

we present adjusted R² values when comparing factors, but raw $R^2$ values when discussing the percentage of variance explained (adjusted $R^2$ values can no longer be interpreted as percent of variance explained). Importantly, we took care to separately filter the QIIME mapping file to exclude 'Unknown' values for each parameter under consideration. Failure to do so can lead to continuous variables (columns containing only numbers) being treated as categorical in QIIME, due to the presence of text values. This in turn can strongly influence inferred R² values.

**Summary of Adonis analysis of microbial beta-diversity**. To present a summarized view of the Adonis analysis of microbial community beta-diversity, we compiled the $R^2$ and p value obtained from each individual Adonis analysis. In Supplementary Fig. 1 the compiled adjusted $R^2$ are presented in a heatmap. In Fig. 1c, we compared the relative influence of each host or environmental parameter on different host compartments by Z-score normalizing Adonis R² values within columns. This has the effect of showing which compartments are most strongly influenced by a particular factor, independent of how influential that factor was overall. We present both views into the data because the unnormalized Adonis R² values better emphasize the absolute magnitude of the microbiome response to each factor, whereas the Z-score normalized values better illustrate common patterns across compartments in host vs. environmental parameters. We emphasize that in both Fig. 1c. and Supplementary Fig. 2, clustering of rows and columns was performed without any prior specification of which factors were host vs. environmental. Thus, observed clustering of host vs. environmental factors emerges from features of the microbial communities themselves.

**Machine learning analyses**. All machine learning analyses were conducted through the supervised_classification.py script in QIIME (v1.9)[44]. This script implements random forest classification, which is a machine learning method for supervised classification. We used default parameters, which classify samples using inferred forests of 500 decision trees. We applied random forest classifiers to two tasks: 1) testing whether we can predict if a DNA sample came from coral mucus, tissue, or skeleton using microbial 16S rRNA data alone and 2) predicting whether within each coral compartment we can predict certain categorical features of a sampled coral using its microbiome (contact with turf algae, reef name, complex vs. robust clade membership, etc.). The results from random forest analysis of coral compartments are presented in the main text. The results for random forest analysis of host and environmental parameters are presented in Supplementary Data S6. Because error rates typically scale with both the number of categories—it is easier to predict the correct category for a binary category than one with 100 possibilities for example—we took care to consider the proportional increase in random classification relative to a baseline formed by random guessing. For both the compartment classification task and the trait classification task, we tested random forest classification on microbial phyla, orders or genera. For the compartment classification task we also tested random forest classification with 97% OTUs directly or predicted functional repertoires of coral microbiomes as inferred using the PICRUSt software. However, this was computationally expensive and yielded<0.1% improvement in classification error rates over classification based on microbial genera, and was therefore not pursued further.

**Statistical analyses on the effect of phylogeny on the microbiome**. Phylogenetic analyses were conducted in R v3.3.1[54]. The packages *ape* (v3.5)[55] and *paleotree* (v2.7)[56] were used to manipulate trees and to calculate cophenetic distances. Univariate phylogenetic correlograms of α-diversity and distance-to-centroid measures were implemented using the package *phylosignal* (v1.1)[57]. Mantel tests and mantel correlograms of multivariate dissimilarities vs. phylogenetic distances were implemented using *vegan* (v2.5–2)[58]. Size classes for the mantel correlograms were defined manually. Following Sturge's rule, we created 11 distance classes. Due to the structure of the host phylogeny and the sampled species, four discrete phylogenetic distances greater than ~0.3 existed, corresponding to (1) all comparisons between the two major coral clades, (2) all comparisons between scleractinians and *Palythoa* (Zoantharia), (3) all comparisons between hexacorals and octocorals, and (4) all comparisons between anthozoans and *Millepora* (Hydrozoa). We first created distance classes that corresponded to each of these four discrete comparisons, then created the remaining seven distance classes by spacing them evenly across the smaller phylogenetic distances.

Phylogenetic Generalized Linear Mixed Models (pGLMMs) for analysis of the entire community were implemented using the package *MCMC.OTU* (1.0.10)[59]. *MCMC.OTU* wraps the package *MCMCglmm* (v2.24)[60] to fit a model whereby sequencing depth is accounted for by using a sample's total read count as the base level of a fixed per-OTU effect. Compositionality is accounted for by the inclusion of a per-sample random effect, and a number of other parameters and priors are set with defaults that are sensible for microbiome studies, such as 'global effects' of each specified factor (which control for and test effects on α-diversity) and independent error variance for each OTU. Analysis of the entire set of 97% OTUs was computationally impractical, so OTUs were first collapsed from the pre-rarefaction OTU table into their annotated genera using QIIME's summarize_taxonomy.py. The package *phyloseq* (1.18.1)[61] was then used to import and manipulate this table and its associated metadata. Samples with total counts less than 1000 or that were lacking relevant metadata were removed. The purgeOutliers command was applied to the data with an otu.cut value of 0.0001.

For the first, more comprehensive GLMM, the command mcmc.otu was run with maximum corallite width, disease prevalence, and binary turf contact as fixed effects; geographic area, host phylogeny, and host identity as random effects; a chain length of 1,25,000, thinning interval of 5, and burn-in of 25,000; and with the inverse of the host phylogenetic covariance matrix supplied with the ginverse option. Subsequently, the command was run again with latitude and then coral colony size as the sole fixed effect, and with only host phylogeny as a random effect. Significance for each term was determined by calculating 95% credible intervals with HPDinterval and isolating those that did not include zero.

**Cophylogenetic analyses**. We reasoned that microbial groups that are most intimately associated with corals (whether commensal, mutualistic, or parasitic) are likely to have evolved in ways that led to patterns of cophylogeny with their hosts. A preliminary pipeline was developed to screen the microbiome for such groups. First, joined sequences were re-processed with the Minimum Entropy Decomposition (MED) pipeline[34], discarding MED nodes with substantive abundances less than 100. Taxonomy was assigned to the resulting MED representative sequences as before with OTUs. Family-level groups of microbes were analyzed independently because higher taxonomic levels would be unlikely to have evolved within the same timescale as scleractinians, and lower taxonomic levels were more likely to contain misannotations. It was computationally impractical to analyze all microbial families, so only the most prevalent in each compartment were tested. An arbitrary threshold of 50% prevalence was chosen. For each family in each compartment, all MED nodes were isolated that were the most abundant representative in at least one sample. This conservative approach was done partly out of concern that spurious sequences generated by sequencing error could influence the downstream phylogenetic analyses, and partly to reduce each dataset to a size that was practical for phylogenetic inference and GLMMs. The representative sequences were then combined from these nodes with reference sequences for each family. Reference sequences were randomly subsampled from the Greengenes 13_8 99% OTU database such that each dataset contained the MED nodes of interest, 75 random full-length 16Ssequences belonging to the family of interest, plus 10 random 'outgroup' sequences belonging to any other family from the same order.

Each collection of sequences was then aligned using MAFFT in QIIME. Phylogenetic trees were built using BEAST 2.4.2 with a chain length of 100 million, thinning interval of 1000, a log-normal relaxed clock model, and the site model selected using bModelTest. The maximum clade credibility tree for each group was selected using TreeAnnotator with a burn-in of 25% and common ancestor heights.

A separate pGLMM was then fit for each microbial family in each tissue compartment. The raw MED table was imported into R using *phyloseq* and filtered to contain only samples with counts greater than 1000. The resulting table was merged with each microbial family's phylogenetic tree using *phyloseq*, a process that automatically filters all sequences from the table that are not represented on the tree. Samples were further filtered from this table if they did not retain a count of at least 10. Phylogenetic covariance matrices based on the bacterial and host phylogenies were then generated[62]. Phylogenetic covariance matrices based on the bacterial and host phylogenies were generated using the function inverseA on each host tree. The Kronecker product of the resulting matrices was then computed for use as the 'coevolutionary' covariance matrix. The Kronecker product of each phylogenetic covariance matrix and an identity matrix was computed for use as microbial identity x host phylogeny and microbial phylogeny × host identity interaction effects[62]

Binary models were fit with *MCMCglmm* using a single fixed effect of the log of the sequencing depth, 'global' random effects of host phylogeny, host identity, microbial phylogeny, and microbial identity, all combinations of host-by-microbe phylogenetic and identity random interaction effects, and a geographic area-by-microbial identity random interaction effect. Altogether this approach is similar to the models described in reference[62]. Our models were fit with a chain length of 1,250,000, thinning interval of 50, and burn-in of 250,000. After the model was fit, convergence was assessed by verifying that the Effective Sample Sizes (ESS) of all covariance terms were greater than 200. Intraclass correlation coefficients (ICCs) were calculated for each iteration, with 95% credible intervals calculated with HPDinterval. Factors with ICC lower credible bounds greater than 0.01 were considered significant.

To independently analyze subclades of *Endozoicomonas*-like bacteria, a custom QIIME-formatted taxonomy database was created with sequence annotations corresponding to clades C, HS, HS-R, and HS-C from the initial analysis. Taxonomy was then assigned to all MED nodes and *Endozoicimonaceae* Greengenes reference sequences using UCLUST with max-accepts set to 1. The above procedure of filtering, selecting reference sequences, building a phylogeny, and fitting pGLMMs, was then repeated based on each annotated subclade instead of each family. HS-R within only Robust clade corals was also analyzed by first filtering other samples from the dataset.

**Code availability**. Analysis code is available on GitHub: https://github.com/zaneveld/GCMP_Australia_Coevolution

## Data availability

Raw sequence data, metadata, OTU and MED representative sequences are publicly available at https://doi.org/10.6084/m9.figshare.c.3855466.v2. Raw sequence data are also deposited at the European Nucleotide Archive under accession number PRJEB28183.

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

## Acknowledgements

The authors would like to acknowledge Tasman Douglass, Margaux Hine, Frazer McGregor, Kathy Morrow, Katia Nicolet, Cathie Page, and Gergely Torda for their field assistance and Lyndsy Gazda, Jamie Lee Proffitt, Gabriele Swain, and Alaina Weinheimer for their assistance in the laboratory. The authors also acknowledge the staff of the Coral Bay Research Station, Lizard Island Research Station, Lord Howe Island Marine Park, Lord Howe Island Research Station, and RV Cape Ferguson for their logistical support. This work was supported by a National Science Foundation Dimensions of Biodiversity grant (#1442306) to R.V.T. and M.M.

## Author contributions

R.V.T., J.Z. and M.M. designed the experiment. F.J.P., R.M., R.V.T. and M.M. conducted the fieldwork. R.M., F.J.P. and S.S. generated the libraries. F.J.P., R.M. and J.Z. performed the data analysis. B.W. and D.B. assisted in the sample collection and provided critical data for meta-analysis. F.J.P., R.M., J.Z., M.M. and R.V.T. wrote the manuscript, and all authors contributed to its editing.

## Additional information

**Competing interests:** The authors declare no competing interests.

