## [Peer Review File · Nature Communications]

Reviewers' Comments:

Reviewer #1:

Remarks to the Author:

This manuscript surveys the evolutionary patterns of bacterial and archaeal richness and composition among corals collected from Australia. The authors find microbial richness and composition to be constrained to some extent by coral phylogeny, with considerable influences from coral life-history strategy and coral compartments. Phylogenetic signal is particularly strong for the host distribution and relationships among Endozoicomonas-like bacteria, suggesting codiversification.

The study uses a rich set of corals, microbial 16S libraries and analyses to explore phylogenetic patterns and test four primary hypotheses relating to microbial communities, host identities, phylogeny and traits. The findings are compelling, novel and well explained. Nevertheless, there remain some doubts that require clarification, and I detail them in the following.

The first pertains to its organisation. The way the hypotheses are listed is rather intuitive, and it would be much easier to follow, considering the rich set of analyses put in, if the analyses are described in the same sequence. The hypotheses go from phyllosymbiosis to cophylogeny/codiversification to disease susceptibility to the more fundamental trait relationships. The results and discussion ought to flow in the same manner. I think it simply can be fixed by moving the two sections on coral disease susceptibility and fast-growing corals to the last part of the results.

An updated (or precise) understanding of the coral phylogeny is needed in many parts of the manuscript. For instance, (1) Pocillopora/Seriatopora/Stylophora-group is simply Pocilloporidae. (2) There are more than 1500 species of Scleractinia (check WoRMS; or add numbers from Cairns 2009, Phylogenetic list of 722 valid recent azooxanthellate scleractinian species, and Carpenter et al. 2008; or check the tree in Huang & Roy 2015). (3) Fossils unambiguous to Scleractinia (as understood pre-2011) appeared in mid-Triassic, but the diversification has in many recent studies been shown to stretch back to the Palaeozoic, with the origin in Ordovician (Stolarski et al. 2011; Huang & Roy 2015; Kitahara et al. 2016). (4) Corals are not the closest relative of bilaterian animals, but rather, some of the closest relatives of bilaterian animals. (5) In line 598, the Fukami et al. (2008) is not recent anymore. There are also issues with those clades being made equivalent to families. See Kitahara et al. (2010, 2016) for family-level groupings.

It is worth noting that this manuscript is focused on reef corals, which are a polyphyletic subset of Scleractinia. More than an issue of taxonomy, the phylogenetic patterns obtained here need to be qualified—essentially the codiversification inferred here does not account for many groups that are either not sampled on the reef, or are deep. This issue can be partially resolved by looking at subgroups, e.g. Acroporidae, which are all reef, or other equivalent clades. Otherwise, we are really just looking at phylogenetic patterns (e.g. cophylogeny), and not processes, which codiversification is. Incomplete sampling on the phylogeny is a perennial issue, and is fine, but also worth a note.

Line 57: Since microbiomes also include other taxa such as dinoflagellates, other protists and fungi, please state why these have not been analysed.

Line 96: The long history has produced differences and convergences. The latter is particularly important for corals, as the disease susceptibility analysis here shows.

Line 102: Is ref #24 more suitable here?

Line 106: Is ref #25 more correct here?

Line 110: Clarify that these are four non-mutually exclusive hypotheses.

Line 188: In Supplementary Data 6, the headers are missing for the last 3 columns.

Line 215: In this and a few other instances later on, the uses of composition, richness and abundance are too loose. In this particular case, looking at Fig. 2b, the authors are really referring to richness and not composition. In another example, in line 247, abundance is mentioned, but I don't see this being shown or tested.

Line 276: Fig. 4 only shows results for tissue, but skeleton microbes are also strongly associated with coral life-history strategy. Perhaps a similar visualisation can be shown as supplementary material.

Line 328: Moran's I is used here because the data are only pertaining to richness? Mantel is used for the microbial community data above. It may be useful to state the difference.

Line 335: The phylogeny in ref #39 is not densely sampled enough to produce these estimates. In fact, the range is much larger, and pushing past 100 Ma in most cases (Stolarski et al. 2011; Huang & Roy 2015).

Line 336: As mentioned above, codiversification is difficult to infer based on incomplete phylogenies. Here, another major issue to highlight is that the conservation of microbial richness/composition on the phylogeny may not be mediated by history, but rather traits (as the paper has also shown) that are phylogenetically constrained. Particularly for associated microbes, it can be difficult to say that the associations have been constrained along lineages which are this old. Unless a dating exercise is also carried out and consistent for microbes, the phylogenetic associations remain as patterns, and changes in microbial richness/composition with coral history/speciation remain elusive.

Line 351: Is this phylogeny the same as the one built in line 166? It would help to streamline the description of the phylogenetic reconstruction.

Line 393: It is unclear why the chloroplast sequences from unicellular eukaryotes are included here. Why were they omitted before? Why include only here?

Line 430: Probably clearer to use 'Generalist' rather than 'Cosmopolitan'.

Line 450: I think 'specialized' is misleading here. HS-C is supposed to be a generalist ('cosmopolitan' by the authors' definition) clade.

Line 504: The fig wasp example may not be equivalent to the cophylogeny between corals and microbes. Do the fig wasps show just host fidelity or phylogenetic associations as well? This is unclear.

Line 536: These algae showing cophylogenetic patterns with corals and outgroups have not been described prior to this. What is this referring to?

Line 549: To be clear, instead of 'diversity', perhaps 'richness'.

Line 699: Are these species not in Huang and Roy 2015 the outgroups? That study appears to have all scleractinians. It's also worth indicating the species (based on the 12S identification, for matching to the phylogeny) in Supplementary Data S1 because some species have been moved among genera (ref

#59).

Line 710: Is there a threshold similarity below which the identification is deemed unreliable? As mentioned by the authors earlier, these sequences can mostly match to genera, so it's unclear how these are used to confirm species identifications.

Line 722: In determining the most abundant mitochondrial type, why is this considered the 12S genotype? Is 12S always the most abundant, or are the authors just matching the names for 12S to the most abundant mitotype?

Line 724: What would branch lengths mean in the relaxed clock analysis? Are there fossil calibrations or mean substitution rates to normalise to time?

Line 740: This is confusing. The analysis using BEAST should return rooted trees. Since there is a strong prior based on numerous past studies that Scleractinia is monophyletic, the outgroups should be (pre)defined. The distinct evolutionary rate between robust and complex corals would almost certainly bias the outgroup position if the outgroup prior is not incorporated (see Kitahara et al. 2014).

Line 742: Why 16S? I thought 12S was used to reconstruct the phylogeny?

Danwei Huang

Reviewer #2:

Remarks to the Author:

I found this manuscript very difficult to read and interpret. This is true in terms of both the objectives/hypotheses and the methodology. The manuscript seems to want to answer every possible question about corals and their microbiome in one go.

Although I found it difficult and convoluted there is no doubt there are some interesting results among the overabundance of tests. As far as I can see the main results are that the microbiomes differ between different coral parts and that they differ in response to the environment and host. I think this is an interesting, clear and important result that should be able to shine through – but in the current manuscript it certainly does not.

I think the authors need to take a step back and reframe this manuscript in terms of clear hypotheses that are carried through the narrative of the paper – even more important is to streamline (and fully explain) the data and data analyses. Until this is done it is impossible to determine whether this work is sound and publishable for Nature Communication.

From the outset it is not clear what the specific aims of the paper are – there are hypotheses set out but on close investigation of them they are very general, lack clarity and don't seem to be independent of each other (i.e. the overlap between phylogenetic signal and coevolution)

Sequencing and phylogenetic analyses: I think the authors sequence both coral and bacteria, but it is very difficult to be sure. The treatment of these sequences is also not clear – the authors seem to have tried to infer a tree from their sequences but find that it does not conform to other coral trees, so they fix the topology to their expectation! This needs explanation – I could imagine a scenario where this would be acceptable but there is no useful explanation here.

Data analyses: I think it is not an exaggeration to say that none of the statistical methodologies in this paper are explained adequately. In addition the overarching approach is not clear either. There are phylogenetic results presented alongside non phylogenetic ones as if they are of equal merit. They are not. They use a variety of packages that essentially do the same thing. There is no useful explanation of why they treat the data the way they do. For example, they use a Mantel test – but it is not clear why or on what data. Why do they not just incorporate depth as a random effect in a GLMM rather than using the convoluted and badly explained rarefaction process? I think the authors could do almost all of their analyses using a single phylogenetic GLMM package (eg MCMCglmm).

I am sorry I can't be more positive at this time. However, I hope the authors get the opportunity to resubmit a reworked, more focused manuscript which I would be happy to review again. However, until then I cannot consider recommending publication.

Reviewer #3:

Remarks to the Author:

This is an original contribution examining the impact of coral compartment, phylogeny, life history and disease susceptibility on the coral microbiome. The analyses are very thoughtful and provide novel insights into unanswered questions about coral microbiomes, and especially the influence of host phylogeny. The paper is a bit of a doozy to thoroughly read through, but it really speaks to the power that one dataset can provide in addressing questions about a particular system. Our community of coral scientists, as well as those studying other microbial habitats, will certainly appreciate the comprehensiveness of the analyses. The biggest weakness of the study is that it was conducted on a small number of sequences (1000) per sample. However, I believe the authors have addressed this weakness by conducting analyses on the samples that were sequenced to a greater depth, which generally show the same results as the 1000 sequences/sample findings. Otherwise, I wish to applaud the authors on producing a fantastic study. I offer relatively minor points below that are intended to strengthen and clarify aspects of the manuscript.

Specific points

1. General note - It is only appropriate to use the term 'bacteria' in your study if you are conducting analysis that only considers the bacterial sequence reads (e.g., *Endozoicomonas* analyses). I suggest that you examine each use of the term 'bacteria' throughout the text to determine if it is appropriately used.
2. General note - Ask the Editor to clarify Nature Comm's guidelines, but in most microbiology journals the families are generally not italicized, only the genus and species.
3. Line 79 – Reference 41 is out of order
4. Line 91- this doesn't read quite right; maybe replace 'testing' with 'to test'
5. Line 125 – In an introduction, my preference is to leave out experimental details, such as the specific software packages
6. Lines 127-129 – This reads rather arrogant and doesn't really reveal much about what you found. I suggest summarizing your results, similarly as you did very nicely in the Abstract
7. Line 131 – Include that this was done in Australia

8. Lines 208 – 212 – This is discussion material.
9. Line 365 (and below) – Not all bacteria (and archaea) are characterized to Genera, so something like, 'microbial taxa' is a better term.
10. Line 370, "Candidatus Amoebophilus", with Candidatus in italics, is the correct nomenclature – also please correct in the Supplementary Results
11. Line 494-495 – This is a confusing statement; do you mean host-specific microbes?
12. Line 640 – Did the rinsing with sterile seawater remove mucus? If so, you might want to mention that.
13. Line 641 – What psi was used to airbrush the tissues? Was the tissue slurry centrifuged and the pellet frozen, or was the entire slurry frozen?
14. Line 663 – What are the units on the DNA concentration?
15. Line 667 – Please check the primer sequences – that is not the 806 revised primer which you cited earlier in the manuscript as using.
16. Line 866 – Change '16S sequence data' to '16S rRNA gene sequence data'
17. Line 866 – Does richness here also include archaea, or it is just bacteria considered?
18. Line 874 – In the legend, please explain the dendrogram
19. Line 888 (legend) – What are the r^2 and p-values of these correlations? I know they are listed in the text, but the subsequent figure has them in the legend, so just be consistent.
20. Line 901 (legend) – Legends should generally 'stand alone' from the text, so can you remind us what GBR & GLMM refer to?
21. Line 910 – Indicate here that the b-e panels are taken from the a plot and have the same axes labels
22. Figure 5 – For the x-axis, since these are all bacteria, you could change to 'Bacterial family'
23. Line 917 – is 'Coral genera' supposed to be 'Microbial family'?
24. Line 947 – For a book, I believe you are supposed to cite the chapter and/or pages.
25. Within references, italicize the coral and bacterial species (lines 954, 958, 1003, etc...)
26. Supplementary files – Please list the File number (S1, S2, ect) at the top of each excel file.

Response to reviewer comments (NCOMMS-17-31961):

Dear Dr. McKay and Reviewers,

Thank you for your insights into our work and for giving us the opportunity to improve this manuscript. We have carefully revised the manuscript to address your comments. In particular, we made several substantial changes to the manuscript to clarify the results and to better emphasize the core findings of our work:

1. We have significantly streamlined the manuscript to better emphasize associations between host phylogeny and the coral microbiome. One reviewer felt strongly that our past submission was attempting to accomplish too many things and requested that we reduce the scope of the work. To address this, we have altered the manuscript significantly and removed two facets of the original paper: 1) correlations between microbial diversity and host disease susceptibility and 2) correlations between microbial community structure and life history traits. This resulted in the removal of two main text figures. This approach allowed us to streamline the paper, improved the flow of the writing, and allowed the manuscript to best emphasize our results related to the questions of phylosymbiosis and co-diversification in the coral microbiome. We believe the remaining aspects of this work are still highly novel and can stand scientifically on their own. Specifically, the question of whether microbial diversity follows coral diversity has not been previously addressed in a comprehensive way, nor have these phylosymbiotic patterns been compared across coral anatomy. Our new modelling approaches incorporates environmental differences between reefs, coral phylogeny, and aspects of coral physiology into a unified model. We think that refocusing the manuscript in this way better emphasizes the main findings that were complemented by reviewers. Lastly, the removal of some data provided us space to more fully interpret and contextualize the biological meaning of a majority of the data.
2. We have added two new figures to our supplemental results section. First, we added a figure describing our analytical pipeline. A major criticism of the work was that the methods to analyze and present the data in each of the figures were not sufficiently clear. We have made a new supplemental figure that diagrams what data and methods are used in each of the analyses and we elaborate on our analytical approaches in the text more completely. We hope this conveys our analytical pipeline more clearly. Second, to best convey multiple perspectives into the data in Fig 1c, we have also added a Supplementary Figure 2, which is identical to Fig. 1c in the main text, except that it presents raw R^2 values, rather than R^2 values that are Z-score normalized within each factor. This highlights host and environmental parameters that have the strongest influence on the microbiome in absolute terms, whereas Fig 1c. highlights differences in the influence of parameters across compartments. Additionally, during our reanalysis, we identified and corrected an error resulting from the QIIME software incorrectly handling continuous variables in Adonis (under certain circumstances they were treated as categorical) and updated Fig. 1c accordingly. This new Fig. 1c is only slightly different than the original and importantly the conclusions remain the same.
3. Removal of results derived from chloroplast data. In order to streamline the work, we have removed the small section on the analysis of chloroplast data from the paper. As there was very little of this in the paper it only changed 4 points on a single figure.

Below, you will find our detailed responses to each comments and suggestions put forward by the reviewers and yourself in a table format. Please note that since we conducted a major reorganization of the text, we have not included a manuscript with visible track changes included as this format would be unreadable.

Editor comments	Author response
Your manuscript entitled "Coral microbiomes reflect host phylogeny, life-history strategy, and disease susceptibility" has now been seen by three referees. I apologize for the delay in delivering this decision to you; several of the reviewers requested multiple extensions. You will see from the reviewer comments below that, while they find your work of interest, some important points are raised. We are interested in the possibility of publishing your study in Nature Communications, but would like to consider your response to these concerns in the form of a revised manuscript before we make a final decision on publication. We therefore invite you to revise and resubmit your manuscript, taking into account the points raised. Please highlight all changes in the manuscript text file.	Thank you for taking the time and effort to critically evaluate our manuscript. We have taken an extended time to reconsider and reformulate this manuscript. The reviewers' comments were constructive and we have taken care to address each of them. We have now significantly modified the scope of the manuscript to address reviewer 2's comments (as we discussed by e-mail). Please note that since we conducted such a major revision, we have not included a manuscript with visible track changes included as this format would be unreadable.

Reviewer 1 comments	Author response
This manuscript surveys the evolutionary patterns of bacterial and archaeal richness and composition among corals collected from Australia. The authors find microbial richness and composition to be constrained to some extent by coral phylogeny, with considerable influences from coral life-history strategy and coral compartments. Phylogenetic signal is particularly strong for the host distribution and relationships among Endozoicomonas-like bacteria, suggesting codiversification. The study uses a rich set of corals, microbial 16S libraries and analyses to explore phylogenetic patterns and test four primary hypotheses relating to microbial communities, host identities, phylogeny and traits. The findings are compelling, novel and well explained. Nevertheless, there remain some doubts that require clarification, and I detail them in the following.	We thank the author for their supportive remarks.
The first pertains to its organisation. The way the hypotheses are listed is rather intuitive, and it would be much easier to follow, considering the rich set of analyses put in, if the analyses are described in the same sequence. The hypotheses go from phyllosymbiosis to cophylogeny/codiversification to disease susceptibility to the more fundamental trait relationships. The results and discussion ought to flow in the same manner. I think it	As mentioned in our above summary, we have significantly altered the paper in an effort to streamline the work and better emphasize our analysis of phyllosymbiosis. As you recommended we have reorganized the Results paragraphs. However, in part as a response to Reviewer 2, we have also decided to remove mention of the four explicit hypotheses from the introduction, opting instead to explain the overall goals

simply can be fixed by moving the two sections on coral disease susceptibility and fast-growing corals to the last part of the results.	of the project more generally. The specific hypotheses did not encompass the entirety of our analysis and may have contributed to confusion about our overall approach.
An updated (or precise) understanding of the coral phylogeny is needed in many parts of the manuscript. For instance, (1) Pocillopora/Seriatopora/Stylophora-group is simply Pocilloporidae.	As recommended we have changed all references to this taxon in the manuscript.
(2) There are more than 1500 species of Scleractinia (check WoRMS; or add numbers from Cairns 2009, Phylogenetic list of 722 valid recent azooxanthellate scleractinian species, and Carpenter et al. 2008; or check the tree in Huang & Roy 2015).	Thank you for pointing this out. We have now updated the number to reflect the current status of WoRMS and added the citation.
(3) Fossils unambiguous to Scleractinia (as understood pre-2011) appeared in mid-Triassic, but the diversification has in many recent studies been shown to stretch back to the Palaeozoic, with the origin in Ordovician (Stolarski et al. 2011; Huang & Roy 2015; Kitahara et al. 2016).	We have reworded references to the origin of corals to emphasize the older age estimates.
(4) Corals are not the closest relative of bilaterian animals, but rather, some of the closest relatives of bilaterian animals.	Very good point. We have changed the wording of this sentence.
(5) In line 598, the Fukami et al. (2008) is not recent anymore. There are also issues with those clades being made equivalent to families. See Kitahara et al. (2010, 2016) for family-level groupings.	We have reworded this sentence to define how recent this phylogeny is. However, the sentence as a whole is important because the phylogeny and clade identities from the Fukami (2008) paper were instrumental in selecting target species during our collections at the beginning of the project. Not all numbered clades from that paper have been made equivalent to families (for example, clade ‘XI’ contains the Fungiidae, but also Coscinaraeidae and Psammocoridae), although we did already reference their approximate equivalence in the following sentence.
It is worth noting that this manuscript is focused on reef corals, which are a polyphyletic subset of Scleractinia. More than an issue of taxonomy, the phylogenetic patterns obtained here need to be qualified—essentially the codiversification inferred here does not account for many groups that are either not sampled on the reef, or are deep. This issue can be partially resolved by looking at subgroups, e.g. Acroporidae, which are all reef, or other equivalent clades. Otherwise, we are really just looking at phylogenetic patterns (e.g. cophylogeny), and not processes, which codiversification is. Incomplete sampling on the phylogeny is a perennial issue, and is fine, but also worth a note.	These are good points. We have added to the discussion that the bias for shallow-water, zooxanthellate corals misses the opportunity to test for effects of these important differences. We agree that for a complete picture of coral-microbe coevolution across the scleractinian tree, samples from deep-water corals should be included. However, given the different methods required for collection (e.g. submersibles), and the many methodological differences (e.g. preservation in alcohol) in existing samples, we did emphasize samples accessible to divers in this first study. Although expensive and technically challenging, adding consistently collected deep water corals and their microbes to the data presented here would be an exciting

	direction for future work (and informally if you have good contacts in this area we really would love to include deep water corals). We had also tried in the original manuscript to address the contrast between ‘cophylogeny’ and ‘codiversification’ (five microbial groups had significant cophylogenetic patterns, but we only suggest that the pattern is consistent with codiversification for the Endozoicimonaceae due to the strength of the signal and manual inspection of the patterns). Nevertheless, we acknowledge that the contrast could be made more explicit and have added a sentence to this regard in the discussion: “A greater geographic breadth of samples and representatives of the many azooxanthellate scleractinians will help inform this notion further, and definitive confirmation of codiversification will additionally require a better-resolved and time-calibrated Endozoicomonas phylogeny.”
Line 57: Since microbiomes also include other taxa such as dinoflagellates, other protists and fungi, please state why these have not been analysed.	Ultimately, analyzing these other taxa is one of the goals of the GCMP, using the same samples. We have added a statement to the end of the introduction that clarified that this is an initial analysis, emphasizing that the initial 16S rRNA data target bacteria and (most) archaea. Further analysis of other microbial groups in these samples, especially Symbiodinium , will be an important next step for the project. We have already begun this process but again to streamline this manuscript we only emphasize the bacteria data.
Line 96: The long history has produced differences and convergences. The latter is particularly important for corals, as the disease susceptibility analysis here shows.	Great point. We have added the word ‘convergences’ to the sentence.
Line 102: Is ref #24 more suitable here?	Yes; but as we have extensively edited the paper, this reference has been removed entirely.
Line 106: Is ref #25 more correct here?	Yes; but the sentence itself has now been removed entirely.
Line 110: Clarify that these are four non-mutually exclusive hypotheses.	We have removed this explicit list of hypotheses entirely.
Line 188: In Supplementary Data 6, the headers are missing for the last 3 columns.	Fixed. Thanks.
Line 215: In this and a few other instances later on, the uses of composition, richness and abundance are too loose. In this particular case, looking at Fig. 2b, the authors are really referring to richness and not composition. In another example, in line 247, abundance is mentioned, but I don’t see this being shown or tested.	We have gone over all the mentions of diversity in the paper and ensured that the terms used are more precisely throughout. On the original line 217, Fig 2b was the correct reference for richness, while composition was tested separately as described in the rest of the paragraph. To clarify this

	point we have added a reference to a supplementary data table into the first sentence that mentions the compositional difference.
Line 276: Fig. 4 only shows results for tissue, but skeleton microbes are also strongly associated with coral life-history strategy. Perhaps a similar visualisation can be shown as supplementary material.	This is an excellent idea. However, as discussed in the summary, we have now removed this section from the manuscript. However, we do aim to do this for the following manuscript that will emphasize life history and disease.
Line 328: Moran's I is used here because the data are only pertaining to richness? Mantel is used for the microbial community data above. It may be useful to state the difference.	Correct. Moran's I is a measure of univariate autocorrelation whereas Mantel tests are used for multivariate data. The exact software and algorithms used to analyze each of them are different and both are explained in the methods, but their theory and interpretation should be similar. We have changed the manuscript to emphasize that they are not identical analyses.
Line 335: The phylogeny in ref #39 is not densely sampled enough to produce these estimates. In fact, the range is much larger, and pushing past 100 Ma in most cases (Stolarski et al. 2011; Huang & Roy 2015).	We have removed reference here to ref #39 due to these limitations. The temporal range mentioned here does, however, seem to correspond to the radiation of most modern reef building families in Huang & Roy 2015. We have reworded this sentence to make this caveat explicit.
Line 336: As mentioned above, codiversification is difficult to infer based on incomplete phylogenies. Here, another major issue to highlight is that the conservation of microbial richness/composition on the phylogeny may not be mediated by history, but rather traits (as the paper has also shown) that are phylogenetically constrained. Particularly for associated microbes, it can be difficult to say that the associations have been constrained along lineages which are this old. Unless a dating exercise is also carried out and consistent for microbes, the phylogenetic associations remain as patterns, and changes in microbial richness/composition with coral history/speciation remain elusive.	We agree that the major patterns of phyllosymbiosis are not likely to be the result of long-term codiversification of the entire microbiome with their hosts. As is stated elsewhere in the paper, it is indeed more likely that certain traits of both hosts and microbes interact to cause them to associate with one another more or less often, and both sets of traits are potentially patterned phylogenetically. It seems reasonable to suggest that emergent effects of these interactions also correspond to ancestral states: a modern genus of corals that is strongly associated with various Gammaproteobacteria is likely to have a common ancestor that itself was susceptible to infection by any number of (potentially different, potentially extinct species of) Gammaproteobacteria, and a modern family of corals that all have low microbiome richness are likely to be descended from an ancestor that itself had a microbiome with low richness. This would be true regardless of the mechanism (large-scale codivergence, which we suggest is not the case, or via influence of host traits). Thus, we stand by our interpretation that “the divergence of coral lineages between roughly 25 and 65 mya was accompanied by large changes in microbiome richness, with changes continuing to accumulate during more recent speciation events.” We have also added a sentence to the discussion that emphasizes that the evidence for codiversification of Endozoicimonaceae is still somewhat speculative until dating exercises are done.

Line 351: Is this phylogeny the same as the one built in line 166? It would help to streamline the description of the phylogenetic reconstruction	We have reworded this sentence to make it clear that we are referring to the same tree, and moved the details introduced here to the previous description of the method.
Line 393: It is unclear why the chloroplast sequences from unicellular eukaryotes are included here. Why were they omitted before? Why include only here?	We have now removed any analysis or discussion of the chloroplast sequences. We hope this removes some of the confusion regarding this minor aspect of the paper.
Line 430: Probably clearer to use ‘Generalist’ rather than ‘Cosmopolitan’.	We have changed references to this clade to ‘Host Generalist’ (abbr. ‘HG’) to avoid conflation with the ‘Generalist’ coral life-history strategy.
Line 450: I think ‘specialized’ is misleading here. HS-C is supposed to be a generalist (‘cosmopolitan’ by the authors’ definition) clade.	Previously, we had named three clades: C, HS-C, and HS-R, with both HS-C and HS-R referring to host specialist groups. We have opted to retain the host specialist abbreviations and change ‘C’ to ‘HG’ (above) as an attempt to make this clearer.
Line 504: The fig wasp example may not be equivalent to the cophylogeny between corals and microbes. Do the fig wasps show just host fidelity or phylogenetic associations as well? This is unclear.	We have removed this example from the manuscript.
Line 536: These algae showing cophylogenetic patterns with corals and outgroups have not been described prior to this. What is this referring to?	We have removed the chloroplast sequences from the manuscript in order to simplify it and focus solely on the Bacterial and Archaeal members of the community.
Line 549: To be clear, instead of ‘diversity’, perhaps ‘richness’.	We agree and have changed it.
Line 699: Are these species not in Huang and Roy 2015 the outgroups? That study appears to have all scleractinians. It’s also worth indicating the species (based on the 12S identification, for matching to the phylogeny) in Supplementary Data S1 because some species have been moved among genera (ref #59).	Indeed, outgroups were among the species missing from the reference tree. Additionally, some of our samples were only identified to the genus level. The 12S sequences provided an empirical way to create the tree given these problems with mapping to the reference. The mapping file included in supplementary data contains columns corresponding to each sample’s 12S sequence variant, field species ID, and ultimate species ID.
Line 710: Is there a threshold similarity below which the identification is deemed unreliable? As mentioned by the authors earlier, these sequences can mostly match to genera, so it’s unclear how these are used to confirm species identifications.	We have reworded the methods so that it is clear that species IDs were not directly confirmed by sequence data / BLAST hits. Rather, it was determined whether field species IDs were reasonable given the sequence data. We followed up on the two samples mentioned that did not appear to have reasonable species IDs by returning to their photos.

Line 722: In determining the most abundant mitochondrial type, why is this considered the 12S genotype? Is 12S always the most abundant, or are the authors just matching the names for 12S to the most abundant mitotype?	All mitochondrial reads were 12S. The presence of multiple unique sequences is almost entirely due to PCR and sequencing error, so it was assumed that the most abundant version is the ‘true’ sequence and the others are generated by sequencing noise. We have clarified this in the manuscript.
Line 724: What would branch lengths mean in the relaxed clock analysis? Are there fossil calibrations or mean substitution rates to normalise to time?	The phylogeny was not time-calibrated, so the branch lengths are only meaningful relative to one another, which should be sufficient for analysis with phylogenetic linear models and other analyses that normalize such estimates beforehand anyway. Without internal time calibration, the relative and ultrametric branch lengths should represent some average between relative divergence times and relative degree of molecular evolution. Thus, analysis using these branch lengths represents a compromise between assuming correlation of traits is proportional to time since divergence and assuming that correlation of traits is proportional to overall evolutionary change since divergence. Discussion of these points has been added to the methods.
Line 740: This is confusing. The analysis using BEAST should return rooted trees. Since there is a strong prior based on numerous past studies that Scleractinia is monophyletic, the outgroups should be (pre)defined. The distinct evolutionary rate between robust and complex corals would almost certainly bias the outgroup position if the outgroup prior is not incorporated (see Kitahara et al. 2014).	Given that the initial phylogenetic reconstruction was mostly performed for quality control purposes, we felt a completely unconstrained inference would be the most informative. The bias in outgroup position did not influence our ability to detect the misidentified individuals, and the strong prior of monophyletic Scleractinia was incorporated into all downstream analyses by fully constraining the subsequent phylogeny to match the Huang and Roy tree. We felt that the entire topology of our low-resolution 12S sequence variants had a defined prior expectation almost as strong as the monophyly of just Scleractinia, and that there was little justification to only constrain some portions of the tree. We have attempted to clarify these points in the methods.
Line 742: Why 16S? I thought 12S was used to reconstruct the phylogeny?	This was a typo; thank you for pointing it out.
Danwei Huang	Thank you for your extremely helpful and constructive comments!

Reviewer 2 comments	Author response
Reviewer #2 (Remarks to the Author): I found this manuscript very difficult to read and interpret. This is true in terms of both the objectives/hypotheses and the methodology. The manuscript seems to want to answer every possible question about corals and their microbiome in one go. Although I found it difficult and convoluted there is no doubt there are some interesting results among the overabundance of tests. As far as I can see the main results are that the microbiomes differ between different coral parts and that they differ in response to the environment and host. I think this is an interesting, clear and important result that should be able to shine through – but in the current manuscript it certainly does not. I think the authors need to take a step back and reframe this manuscript in terms of clear hypotheses that are carried through the narrative of the paper – even more important is to streamline (and fully explain) the data and data analyses. Until this is done it is impossible to determine whether this work is sound and publishable for Nature Communication. From the outset it is not clear what the specific aims of the paper are – there are hypotheses set out but on close investigation of them they are very general, lack clarity and don't seem to be independent of each other (i.e. the overlap between phylogenetic signal and coevolution)	We appreciate your candor and have taken considerable time and care to clarify our overall strategy and the details of our methods. We have significantly revised the manuscript in an effort to do more with less. Given your concerns, and after discussion with the editor, we have removed some aspects of the work to streamline its focus and to relay more completely the novel findings. Specifically, we now focus on the phyllosymbiosis aspects of the work exclusively. We have also removed mention of the specific hypotheses from the introduction in favor of a more general description of the study's goals. We have reorganized the figures and paragraphs of the results section so that they are in better alignment with the discussion and follow a logical progression: we (1) first list a number of basic findings and statistics from the dataset; then (2) assess the influence of the coral phylogeny via overall patterns phyllosymbiosis and specific instances of cophylogeny.
Sequencing and phylogenetic analyses: I think the authors sequence both coral and bacteria, but it is very difficult to be sure. The treatment of these sequences is also not clear – the authors seem to have tried to infer a tree from their sequences but find that it does not conform to other coral trees, so they fix the topology to their expectation! This needs explanation – I could imagine a scenario where this would be acceptable but there is no useful explanation here.	The only coral sequences that we generated were 'off-target' 12S sequences that were in the same '16S' MiSeq sequence libraries as the bacterial and archaeal reads. We have attempted to clarify this in the manuscript. We have also included a discussion of the reasons for and consequences of our methods of construction of the reference-based host phylogeny. In brief, the opportunistic host data were not considered strong enough to construct a de novo topology, but were considered somewhat informative about evolutionary distances and the general placement of some taxa that were missing from the reference.
Data analyses: I think it is not an exaggeration to say that none of the statistical methodologies in this paper are explained adequately. In addition the overarching approach is not clear either. There are phylogenetic	We agree that non-phylogenetic methods are not of equal merit as those that account for phylogeny. Many of the non-phylogenetic methods we used in the paper are essentially field standards and have been used to ask similar questions in the past (for instance, simple

results presented alongside non phylogenetic ones as if they are of equal merit. They are not.	correlations of microbes with host traits), and part of our intent was to show exactly this – how methods that incorporate the phylogenetic history of the host along with other key parameters represent a more robust approach to inferring the drivers of host-microbe associations in a unified statistical framework. We think the point that you have made about phylogenetic vs. non-phylogenetic methods is an important one for the field, and can be illustrated by comparison. We have added statements highlighting the contrast between phylogenetically-informed or non-phylogenetic (i.e. naïve) methods to the manuscript. We have also added additional detail to the methods section and a supplementary workflow figure (Fig. S1) to clarify the statistical methods.
They use a variety of packages that essentially do the same thing. There is no useful explanation of why they treat the data the way they do. For example, they use a Mantel test – but it is not clear why or on what data. Why do they not just incorporate depth as a random effect in a GLMM rather than using the convoluted and badly explained rarefaction process? I think the authors could do almost all of their analyses using a single phylogenetic GLMM package (eg MCMCglmm).	We have rewritten parts of our statistical methods section to clarify the pipeline and to include more detail about the models. In fact, GLMMs were conducted on pre-rarefaction OTU tables and did include sequencing depth as an effect. The package we used for the community-wide analysis does this by default, and the models we cited for the cophylogeny analysis optionally include similar effects as well. We have now explained these models in more detail and emphasized these points. We have also created a new supplementary figure that diagrams the data pipeline and helps to clarify the steps that led to each analysis and figure.
I am sorry I can't be more positive at this time. However, I hope the authors get the opportunity to resubmit a reworked, more focused manuscript which I would be happy to review again. However, until then I cannot consider recommending publication.	We have taken your comments very seriously and revised this work significantly. We hope the new resubmission better presents our work and convinces you that the data are novel and worthy of publication

Reviewer 3 comments	Author response
This is an original contribution examining the impact of coral compartment, phylogeny, life history and disease susceptibility on the coral microbiome. The analyses are very thoughtful and provide novel insights into unanswered questions about coral microbiomes, and especially the influence of host phylogeny. The paper is a bit of a doozy to thoroughly read through, but it really speaks to the power that one dataset can provide in addressing questions about a particular system. Our community of coral scientists, as well as those studying other microbial habitats, will certainly appreciate the comprehensiveness of the analyses. The biggest weakness of the study is that it was conducted on a small number of sequences (1000) per sample. However, I believe the authors have addressed this weakness by conducting analyses on the samples that were sequenced to a greater depth, which generally show the same results as the 1000 sequences/sample findings. Otherwise, I wish to applaud the authors on producing a fantastic study. I offer relatively minor points below that are intended to strengthen and clarify aspects of the manuscript.	We appreciate the reviewer's kind words, as well as the useful suggestions noted below.
The paper is a bit of a doozy to thoroughly read through, but it really speaks to the power that one dataset can provide in addressing questions about a particular system.	We agree that the previous draft was too broad and at times tough to read. Based on your comments and those of reviewer 2, and after consultation with the editor, we have decided to reduce the scope of the results somewhat to emphasize the core conclusions about phylosymbiosis. Hopefully this streamlines the paper and makes it more accessible.
1. General note - It is only appropriate to use the term 'bacteria' in your study if you are conducting analysis that only considers the bacterial sequence reads (e.g., Endozoicomonas analyses). I suggest that you examine each use of the term 'bacteria' throughout the text to determine if it is appropriately used.	We have gone through the manuscript and ensured that the terms 'bacteria' and 'microbes' are used appropriately.
2. General note - Ask the Editor to clarify Nature Comm's guidelines, but in most microbiology journals the families are generally not italicized, only the genus and species.	We have removed italics from higher-level taxonomic names, as is standard in British journals.
3. Line 79 – Reference 41 is out of order	Fixed.
4. Line 91- this doesn't read quite right; maybe replace 'testing' with 'to test'	Fixed.
5. Line 125 – In an introduction, my preference is to leave out experimental details, such as the specific software packages	We have removed reference to the specific packages.
6. Lines 127-129 – This reads rather arrogant and doesn't really reveal much about what you found. I suggest summarizing your results, similarly as you did very nicely in the Abstract	Thanks for this catch! We can see how this came off as arrogant, and have removed the statement.

	The results did in fact change our understanding, but may or may not do much for the wider community! In particular there were a lot of hypotheses that we had thought were reasonable going in to the project, and wrote down in the grant that funded it, but that turned out to be totally contradicted by the data. For example, we thought that mucus microbiomes would better predict disease susceptibility than tissue microbiomes, and that coral skeleton would be less diverse than mucus.
7. Line 131 – Include that this was done in Australia	Done.
8. Lines 208 – 212 – This is discussion material.	It has been removed to streamline the paper.
9. Line 365 (and below) – Not all bacteria (and archaea) are characterized to Genera, so something like, ‘microbial taxa’ is a better term.	These analyses were conducted on data that had been collapsed to microbial genera as defined in GreenGenes, and as such we feel that it is best to maintain this more precise wording. We have, however, clarified in the discussion that some ‘genera’ represent imprecise pools of unresolved taxa.
10. Line 370, “Candidatus Amoebophilus”, with Candidatus in italics, is the correct nomenclature – also please correct in the Supplementary Results	Done, thank you.
11. Line 494-495 – This is a confusing statement; do you mean host-specific microbes?	Yes, and we have clarified the statement as ‘host-specific microbial genera’.
12. Line 640 – Did the rinsing with sterile seawater remove mucus? If so, you might want to mention that.	Yes, and we have added this information to methods.
13. Line 641 – What psi was used to airbrush the tissues? Was the tissue slurry centrifuged and the pellet frozen, or was the entire slurry frozen?	These details have been added to the methods.
14. Line 663 – What are the units on the DNA concentration?	Good catch; we have added units.
15. Line 667 – Please check the primer sequences – that is not the 806 revised primer which you cited earlier in the manuscript as using.	The sequences and names are correct, but the initial citation was incorrect; thank you for pointing this error out.
16. Line 866 – Change ‘16S sequence data’ to ‘16S rRNA gene sequence data’	Fixed.
17. Line 866 – Does richness here also include archaea, or it is just bacteria considered?	It includes archaea. We have ensured that comments about the bacterial & archaeal community are now referred to as ‘microbial’ rather than ‘bacterial’, throughout the manuscript
18. Line 874 – In the legend, please explain the dendrogram	Thank you – we now explain the dendrogram in the figure legend. The dendrograms represent UPGMA clustering of parameters (dendrogram on x-axis) or compartments (dendrogram on y-axis) according to their adjusted Adonis r ² value. So compartments whose microbiomes respond to similar parameters are clustered together on the y-axis, while parameters that influence similar compartments are clustered together on the x-axis. It’s worth emphasizing that the method knows

	nothing about which parameters are ‘host’ or ‘environmental’ yet these still separate as the deepest split on the x-axis dendrogram.
19. Line 888 (legend) – What are the r ² and p-values of these correlations? I know they are listed in the text, but the subsequent figure has them in the legend, so just be consistent.	Thanks for this comment - we have added substantial statistical detail to the figure legends of this figure. We present a simple Pearson regression for visualization, but in keeping with recommendations from Reviewer 2 present effect sizes and p values from phylogenetic Generalized Linear Mixed Model (pGLMM) analysis, as this more appropriately accounts for the uneven distribution of coral species across latitudes.
20. Line 901 (legend) – Legends should generally ‘stand alone’ from the text, so can you remind us what GBR & GLMM refer to?	This is an excellent point, and we agree that some our figure legends were too terse to really support a ‘legends-first’ reading of the paper. We’ve added more detail and explanation to the figure legends to make them more self-contained.
21. Line 910 – Indicate here that the b-e panels are taken from the a plot and have the same axes labels	This is a good suggestion; thank you. However, this figure was removed when streamlining the manuscript along the lines of Reviewer 2’s suggestion. We’ll keep this adjustment in mind if we present the data in a later work.
22. Figure 5 – For the x-axis, since these are all bacteria, you could change to ‘Bacterial family’	This originally included eukaryotic chloroplast sequences, but since their analysis was removed from the manuscript, we will take your suggestion and now only refer to bacterial families!
23. Line 917 – is ‘Coral genera’ supposed to be ‘Microbial family’?	Yes, this has been corrected.
24. Line 947 – For a book, I believe you are supposed to cite the chapter and/or pages.	This reference was replaced with the WoRMS citation
25. Within references, italicize the coral and bacterial species (lines 954, 958, 1003, etc...)	Fixed.
26. Supplementary files – Please list the File number (S1, S2, ect) at the top of each excel file.	Done.

Reviewers' Comments:

Reviewer #1:

Remarks to the Author:

The manuscript is certainly much improved. I applaud the authors for the herculean effort. It is much clearer now, but I probably wouldn't characterise this as a streamlining as the manuscript is even longer now! I'm not too concerned because it is generally a nice read, though shortening the text at some areas may help the general readership.

I emphasise again that there is really no evidence for codiversification. The issue is not just the limitation that the host phylogeny is lacking azooxanthellate corals and in fact most zooxanthellate corals, but also that the timescales are not explicitly tested. Specifically, there is no support for the statement 'The highly significant cophylogeny terms from both these tests (ICCs, 95% lower bounds: 0.34 and 0.21, respectively) are consistent with the long-term codiversification of clade HS at least within the Robust clade of corals, and possibly since the last common ancestor of all Scleractinia.' All this shows is a certain degree of cophylogeny and phylogenetic constraint on the microbial clades by the host phylogeny—these represent just a few nodes on the phylogenies. For the vast majority of nodes on both trees, there are limited co-branching patterns. Without time calibrations on both phylogenies, there is no way to tell if the diversification of both host and Endozoicomonas-like bacteria is occurring at the same time.

Reviewer #2:

Remarks to the Author:

The authors have taken what I said in my previous comments very seriously. I think the revised manuscript is significantly better both in scope and clarity. There are still some issues associated with reporting R^2 and pseudo R^2 values – these values are often very approximate. What would be more useful is the significance of parameters and a different measure of effect size – perhaps the difference in DIC or some such (this could be in addition to the R^2 of the authors really like it) but one should never judge significance on the basis of R^2 .

In addition I cannot support the publication of this manuscript while it still reports phylogenetic results alongside non-phylogenetic results. Doing this on the basis of some field specific culture or history is just not acceptable. If there is phylogenetic signal in the data reporting non-phylogenetic results is fundamentally and scientifically unsound – it is like reporting results from an experiment with enormous amounts of pseudo-replication. Of course, no credible journal would accept that.

We thank all reviewers for their thoughtful and constructive guidance during our revisions.

Reviewers' comments:

Reviewer #1 (Remarks to the Author):

The manuscript is certainly much improved. I applaud the authors for the herculean effort. It is much clearer now, but I probably wouldn't characterise this as a streamlining as the manuscript is even longer now! I'm not too concerned because it is generally a nice read, though shortening the text at some areas may help the general readership.

Thank you again for suggesting we streamline the paper. We agree that it has improved the study. In this minor revision, we have further revised and shortened the work to better emphasize the core findings of the study.

I emphasise again that there is really no evidence for codiversification. The issue is not just the limitation that the host phylogeny is lacking azooxanthellate corals and in fact most zooxanthellate corals, but also that the timescales are not explicitly tested.

We agree that the data presented in the paper demonstrates lack of codiversification of the entire group of coral-associated Endozoicomonas, and is consistent with scenarios other than codiversification even in the smaller subsets of Endozoicomonas that are monophyletic and cophylogenetic. Our intent in the last draft was not to express that the data definitively prove codiversification but rather to express that they are consistent with it in one Endozoicomonas subgroup. Nevertheless, we have removed all mentions of codiversification from the paper except when listing it as one of many explanations for phylosymbiosis and when explicitly stating that a time-calibrated microbial phylogeny would be necessary to distinguish codiversification from other forms of cophylogeny.

Specifically, there is no support for the statement 'The highly significant cophylogeny terms from both these tests (ICCs, 95% lower bounds: 0.34 and 0.21, respectively) are consistent with the long-term codiversification of clade HS at least within the Robust clade of corals, and possibly since the last common ancestor of all Scleractinia.' All this shows is a certain degree of cophylogeny and phylogenetic constraint on the microbial clades by the host phylogeny—these represent just a few nodes on the phylogenies. For the vast majority of nodes on both trees, there are limited co-branching patterns. Without time calibrations on both phylogenies, there is no way to tell if the diversification of both host and Endozoicomonas-like bacteria is occurring at the same time.

We removed all mention of codiversification from this sentence and in relation to this test.

Reviewer #2 (Remarks to the Author):

The authors have taken what I said in my previous comments very seriously. I think the revised manuscript is significantly better both in scope and clarity.

Thank you for your kind comments about our revision.

There are still some issues associated with reporting R^2 and pseudo R^2 values – these values are often very approximate. What would be more useful is the significance of parameters and a different measure of effect size – perhaps the difference in DIC or some such (this could be in addition to the R^2 of the authors really like it) but one should never judge significance on the basis of R^2 .

We believe you are referring to the R^2 value reported in figure 2, where significance was tested using pGLMMs. We have removed these, and now report pGLMM effect sizes in the legend of Figure 2. We agree that R^2 is not a measure of significance, and instead report either pMCMC (for pGLMM analysis) or the permutational p-value (for Adonis analysis) for significance.

In addition, I cannot support the publication of this manuscript while it still reports phylogenetic results alongside non-phylogenetic results. Doing this on the basis of some field specific culture or history is just not acceptable. If there is phylogenetic signal in the data reporting non-phylogenetic results is fundamentally and scientifically unsound – it is like reporting results from an experiment with enormous amounts of pseudo-replication. Of course, no credible journal would accept that.

We agree. Our initial rationale for taking this approach was to demonstrate why you should use phylogenetically aware methods instead of non-phylogenetic methods when available. However, we recognize that this may have missed the mark and lead to confusion. Thus, we have remade the figures and removed any reference to these comparisons from the text.

** See Nature Research's author and referees' website at www.nature.com/authors for information about policies, services and author benefits

This email has been sent through the Springer Nature Tracking System NY-610A-NPG&MTS

Confidentiality Statement:

This e-mail is confidential and subject to copyright. Any unauthorised use or disclosure of its contents is prohibited. If you have received this email in error please notify our Manuscript Tracking System Helpdesk team at <http://platformsupport.nature.com>.

Details of the confidentiality and pre-publicity policy may be found here <http://www.nature.com/authors/policies/confidentiality.html>

Privacy Policy | Update Profile